# Hillslope and groundwater contributions to streamflow in a Rocky Mountain watershed underlain by glacial till and fractured sedimentary bedrock

Sheena A. Spencer[1], Axel E. Anderson[1,2], Uldis Silins[1], Adrian L. Collins[3]

[1]Department of Renewable Resources, University of Alberta, Edmonton, T6G 2G7, Canada
[2]Alberta Agriculture and Forestry, Government of Alberta, Edmonton, T5K 1E4, Canada
[3]Sustainable Agriculture Sciences, Rothamsted Research, North Wyke, Okehampton, EX20 2SB, United Kingdom

*Correspondence to*: Sheena A. Spencer (sheena.spencer@ualberta.ca)

**Abstract.** Permeable sedimentary bedrock overlain by glacial till leads to large storage capacities and complex subsurface flow pathways in the Canadian Rocky Mountain region. While some inferences on the storage and release of water can be drawn from conceptualizations of runoff generation (e.g., runoff thresholds and hydrologic connectivity) in physically similar watersheds, relatively little research has been conducted in snow-dominated watersheds with multi-layered permeable substrates that are characteristic of the Canadian Rocky Mountains. Stream water and source water (rain, snowmelt, soil water, hillslope groundwater, till groundwater, and bedrock groundwater) were sampled in four sub-watersheds (Star West Lower, Star West Upper, Star East Lower and Star East Upper) in Star Creek, SW Alberta to characterize the spatial and temporal variation in source water contributions to streamflow in upper and lower reaches of this watershed. Principal component analysis was used to determine the relative dominance and timing of source water contributions to streamflow over the 2014 and 2015 hydrologic seasons. An initial displacement of water stored in the hillslope over winter ("reacted water" rather than "unreacted" snowmelt and rainfall) occurred at the onset of snowmelt before stream discharge responded significantly. This was followed by a dilution effect as snowmelt saturated the landscape, recharged groundwater, and connected the hillslopes to the stream. Fall baseflows were dominated by either riparian water or hillslope groundwater in Star West. Conversely, in Star East, the composition of stream water was similar to hillslope water in August but plotted outside the boundary of the measured sources in September and October. The chemical composition of groundwater seeps followed the same temporal trend as stream water, but consistently cold temperatures of the seeps suggested deep groundwater was likely the source of this late fall streamflow. Temperature and chemical signatures of groundwater seeps also suggest highly complex subsurface flow pathways. The insights gained from this research help improve our understanding of the processes by which water is stored and released from watersheds with multi-layered subsurface structures.

# 1 Introduction

Forest disturbance from wildfire, pathogens, or forest harvesting removes the forest canopy increasing the total precipitation that reaches the forest floor (Williams et al., 2019; Burles and Boon, 2011; Boon, 2012; Pugh and Small, 2012; Varhola et al., 2010) often altering the dominant flow pathways, increasing streamflow quantity and changing the timing of flows in forested watersheds (Stednick, 1996; Scott, 1993; Bearup et al., 2014; Winkler et al., 2017). However, large variability has been observed in streamflow responses following disturbance due to differences in disturbance type, vegetation type, precipitation regimes, and soil moisture storage (Brown et al., 2005; Stednick, 1996). Some studies in Alberta's Rocky Mountains have reported little, if any, change in streamflow following disturbance (Williams et al., 2015; Harder et al., 2015; Goodbrand and Anderson, 2016; Andres et al., 1987) but the mechanisms and watershed features (e.g., bedrock, surficial geology, wetlands) potentially responsible for the lack of flow response have received little attention. It has been suggested that watersheds exhibiting a lack of change in streamflow following disturbance might be associated with a large storage capacity and complex subsurface flow pathways (Harder et al., 2015) but the higher-order controls regulating these muted responses remain unclear. Runoff generation has been extensively studied in regions with relatively impermeable bedrock overlain by shallower soils, which has led to broadly accepted conceptualizations of runoff dynamics (e.g., old water contributions to streamflow, macropore flow and subsurface streamflow generation (McGlynn et al., 2002); fill and spill hypothesis (Tromp-van Meerveld and McDonnell, 2006); hillslope-stream connectivity (Jencso et al., 2009)). However, these conceptualizations may not apply to regions with more complex structural controls on runoff such as permeable bedrock, deeper soils, or where multiple subsurface systems interact. Runoff generation in Alberta's Rocky Mountains has added complexity because of the combination of both permeable sedimentary bedrock (highly fractured and faulted) and an overlying layer of deep, heterogenous glacial till (3 m deep on average, up to 10 m deep) (AGS, 2004; Waterline Resources, Inc., 2013). This is in contrast to regions such as the southern Rocky Mountains in Colorado (Sueker et al., 2000; Cowie et al., 2017) and Montana (Jencso et al., 2009; 2010; Nippgen et al., 2015) that are often dominated by less permeable metamorphic or igneous bedrock and thinner soils. While runoff generation processes may differ from these regions, some inferences can be drawn from studies in regions with either permeable bedrock or deep soils alone. Watersheds with high bedrock permeability have been associated with longer subsurface flow pathways and the slow release of stored water to streams during baseflow (Uchida et al., 2006; Liu et al., 2004; Pfister et al., 2017). Uchida et al. (2006) reported that a watershed with greater bedrock permeability had larger aquifer storage, and the subsequent release of stored water maintained baseflow later in the year. Similarly, Liu et al. (2004) showed that the recession limb of the annual hydrograph in the Colorado front range Rocky Mountains was driven by baseflow released from fractured bedrock but Cowie et al. (2017) also stressed the importance of talus slopes as a source of streamflow in the same alpine watershed. Deep soils and till deposits with large storage capacities have also been shown to sustain baseflows during drier periods (Floriancic et al., 2018; Shanley et al., 2015). Deep sediment deposits in the Poschiavino watershed, in Switzerland, were associated with greater storage capacity and higher winter baseflows compared to watersheds with shallow sediment deposits (Floriancic et al., 2018). Similarly, deep basal till in Sleepers River watershed in Vermont was

associated with large storage capacity and low permeability that promoted extended maintenance of baseflow (Shanley et al.,

2015).

While these studies illustrate the influence of permeable or fractured bedrock, deep soils, or till on baseflows, few studies have explored the combination of these storage zones on streamflow contributions (Burns et al., 1998; Dalke et al., 2012; Shaman et al., 2004). Burns et al. (1998) characterized the difference in deep (bedrock) and shallow (soils and till) flow systems in the Catskill Mountains in New York, a region with both glacial till and permeable sedimentary bedrock. Baseflow was maintained

by discharge from perennial springs, which originated from bedrock fractures, rather than contributions from the soil and till flow system (Burns et al., 1998). Conversely, fragipan layers contributed to differing flow systems under dry vs wet antecedent conditions in central New York state, USA (Dalke et al., 2012). Stormflow was generated from deep flow pathways below the fragipan during dry conditions and near surface flow pathways during wet conditions. Comparatively little research on runoff generation processes has been conducted in the Canadian Rocky Mountains in part due to deep snow that is present for much

of the year (Oct-May). While some studies have shown the importance of groundwater contributions to streamflow in alpine watersheds in the Rocky Mountains (Hood and Hayashi, 2015; McClymont et al., 2010), the additional complexity imposed by highly heterogeneous glacial till and permeable bedrock in sub-alpine and upper montane watersheds has limited more extensive research on runoff dynamics of this region. As a first attempt to conceptualize runoff generation in Alberta's Rocky Mountains, Spencer et al. (2019) quantified storage and precipitation-runoff relationships from hydrometric data. Results

indicated that runoff generation was strongly governed by the interaction of two zones of storage – soil and till storage and bedrock storage. The alpine region and sub-alpine/upper montane region were also identified as two separate hydrologic response units that differed in timing and flow pathways for runoff response. While Spencer et al. (2019) developed a conceptualization of runoff generation for this region, they concluded that coupled flow and tracer approaches would be needed to reduce uncertainty in estimated flow contributions from each storage zone.

Chemical signatures of source water (rain, snowmelt, soil water, hillslope groundwater, till groundwater, and bedrock groundwater) and stream water can be used to determine which sources are contributing to streamflow during different flow conditions using end-member mixing analysis (EMMA; Christophersen and Hooper, 1992). The key assumptions for EMMA are: 1) the tracers are conservative; 2) the mixing process is linear; 3) source chemistry does not change temporally or spatially over the period or area studied (Inamdar, 2011; Hooper, 2003), and; 4) all sources have been identified and have the potential

to contribute to streamflow. Many studies have used EMMA to conceptualize when different geologic components are contributing to the stream (e.g., James and Roulet, 2006; Cowie et al., 2017; Ali et al., 2010). However, this approach has been most successful in smaller watersheds (1 km$^2$) because of more constrained variation in source water at smaller spatial scales (Hoeg et al., 2000). Large watersheds could be characterized based on smaller sub-watersheds (James and Roulet, 2006) if sub-watersheds are homogeneous. Others have concluded that where source water displays large variation or assumptions

cannot be met, runoff processes should be described qualitatively (Inamdar et al., 2013; Hoeg et al., 2000; Correa et al., 2019). To expand on recent work carried out in the same study area by Spencer et al. (2019), this study aims to advance the conceptualization of runoff generation in the Rocky Mountains in Alberta, Canada. The objectives of this study were to: 1)

characterize how sources of stream water (rain, snowmelt, soil water, hillslope groundwater, till groundwater, and bedrock groundwater) vary spatially across four sub-watersheds of a Rocky Mountain watershed and temporally from spring snowmelt to the start of the next year's snow accumulation period, and; 2) determine the relative contributions of source water to the stream from spring to fall for each sub-watershed. This study should help inform the current conceptualization of runoff generation in northern Rocky Mountain watersheds.

## 2 Study Site

Star Creek watershed (10.4 km$^2$; Figure 1) is located in the eastern slopes of Canada's Rocky Mountains; a region with fractured sedimentary bedrock overlain by glacial till. Average annual precipitation was 720 mm at Star Main (1482 m a.s.l.) and 990 mm at Star Alpine (1732 m a.s.l.; Spencer et al., 2019). The area-weighted average annual precipitation (2005-2018) was 950 mm using the Thiessen-polygon method and nine precipitation gauges at a range of elevations in and surrounding Star Creek; 50-60 % of the precipitation falls in the form of snow (Spencer et al., 2019). Soils are Eutric Brunisols (Can. Soil Classification, or Eutric Cambisols in Food and Agriculture Organization system) approximately 1 m deep, on average. Star Creek is underlain by unsorted and uncompacted glacial till which is generally less than 3 m deep with an estimated total area of 2.4 km$^2$ (AGS, 2004; Figure 1). Some clay-rich till layers, likely from localized glacial ice melt features, occur intermittently throughout the watershed resulting in heterogeneous and uneven distribution of glacial till throughout the watershed. Sedimentary geologic formations (Upper Paleozoic formation, Belly River-St. Mary Succession, and Alberta Group formation) are primarily composed of shale and sandstone (AGS, 2004) and are highly fractured due to folding and faulting (Waterline Resources, Inc., 2013).

Star Creek includes two main sub-watersheds, Star East (3.9 km$^2$; 1537-2628 m above sea level) and Star West (4.6 km$^2$; 1540-2516 m above sea level). Unvegetated talus slopes (0.50 km$^2$ in Star East and 0.53 km$^2$ in Star West, digitized from orthoimages) and exposed bedrock form the upper portion of alpine zones in both sub-watersheds (Figures 1 and 2). Talus slopes terminate in the alpine and transitional forested regions of the watershed but streams or tributary features flowing from talus slopes have not been observed. There is also no evidence of permafrost, ice lenses, or rock glaciers, unlike in other Rocky Mountain regions (Cowie et al., 2017; Clow et al. 2003; Hood and Hayashi et al., 2015; McClymont et al., 2010). Star West has a larger alpine region with cirque till deposits (estimated area of 0.14 km$^2$ (AGS, 2004)) that includes a narrow marshy area proximal to the stream that holds water throughout the summer and drains into the main channel that is primarily bedrock in the upper reaches. The Star East alpine region is smaller and more constricted than in Star West (Figure 2) and is comprised mostly of a grassy meadow with the stream originating from springs where the water table reaches the soil surface and is incised in colluvium with large boulders. In the lower reaches, streams in both sub-watersheds are composed of a series of step-pools incised in alluvium and colluvium with some areas of exposed bedrock.

Two historical streamflow gauging sites exist in each sub-watershed – a lower site (Star West Lower (SWL) and Star East Lower (SEL)) near the confluence of the two sub-watersheds (1540 m above sea level) and an upper site (Star West Upper

(SWU) and Star East Upper (SEU)) located at approximately 1690 m above sea level in the sub-alpine transition zone (Figure 1). The sub-alpine and upper Montane zones are dominated by sub-alpine fir (*Abies lasiocarpa*) and Englemann spruce (*Picea Englemannii*) above lodgepole pine (*Pinus contorta*) dominated forests at lower elevations (Dixon et al., 2014; Silins et al., 2009). Vegetation in upper and lower watersheds (Figure 1) are distinguished by a transition between higher elevation alpine heath/shrub vegetation and sub-alpine fir dominated forests in the upper watersheds and lodgepole pine dominated forest in 135 the lower watersheds.

## 3 Methods

### 3.1 Stream water chemistry

Stream water samples were collected from the four streamflow gauging stations (SEL, SEU, SWL, SWU; Figure 1) every two weeks from April to October, in 2014 and 2015, to capture the full range in streamflow chemistry over the hydrologically 140 active period. One litre plastic bottles were triple rinsed prior to sample collection. Samples were analyzed for major cations and anions ($Na^+$, $Mg^{2+}$, $Ca^{2+}$, $K^+$, $Cl^-$, $SO_4^{2-}$) and silica (Si as $SiO_2$) in the Biogeochemical Analytical Service Laboratory (University of Alberta). An inductively coupled plasma-optical emission spectrometer (Thermo Scientific ICAP 6300) was used to measure $Na^+$, $Mg^{2+}$, $Ca^{2+}$, $K^+$ with analytical precision of 1.9 %, 3.0 %, 1.9 %, and 2.4 %, respectively. An ion chromatograph (Dionex DX600 and Dionex ICS 2500) was used to measure $Cl^-$ and $SO_4^{2-}$ with analytical precision of 2.4 % 145 and 3.1 %. Flow injection analysis (Lachat QuikChem 8500 FIA automated ion analyzer) was used to measure Si with analytical precision of 3.4 %.

Continuous stream discharge was estimated from stage-discharge relationships developed at each gauging station. Stage was measured at a 10-minute interval using a bubbler system (H350/H355 Waterlog Series, YSI Inc./Xylem Inc., Yellow Spring, Ohio, USA) or a pressure transducer (HOBO U20, Onset Computer Corp., Bourne, MA, USA). Discharge measurements were 150 taken with a velocimeter (SonTek/Xylem Inc., San Diego, CA, USA) 12-18 times from April to October at each site in 2014 and 2015.

### 3.2 Source water chemistry

Stream water sources were a priori hypothesized to consist of rain, snowmelt, soil water, hillslope groundwater, till groundwater, and bedrock groundwater (seeps used as a proxy for till and bedrock groundwater) based on field observations 155 and inferences made from research conducted in this watershed since 2004 (Silins et al., 2016). All source water samples were collected in triple rinsed (with source water) 50 ml plastic vials and analyzed with the same methods as the stream water samples to support application of end-member mixing analysis. Source water collection and sampling methods are detailed below.

### 3.2.1 Rainfall and snowmelt

Rain samples were collected in clean buckets rinsed with deionized water. Buckets were placed in open areas throughout the watershed or in the nearby townsite (Coleman, AB; within approximately 8 km of Star Creek watershed) after a rainstorm began. Locations were chosen opportunistically depending on storm timing and site access. Samples were collected at the end of the day or once there was enough water in the bucket to sample to prevent changes in chemical composition due to dry deposition of dust or evaporation. Five, four, and three samples were collected throughout the summers of 2013, 2014, and

2015, respectively. The difficulty of capturing large convective storms and the large frequency of storms less than 5 mm (Williams et al., 2019) prevented the collection of more rainfall samples.

Nine snowmelt samples were collected from sub-alpine regions of Star Creek and two from North York Creek (an adjacent watershed, Figure 1) throughout spring and early summer in 2014. Three additional samples were collected in spring 2015 but mid-winter melt of snowpacks hindered the collection of more snowmelt samples. Eavestroughs, 3 m in length, were installed

perpendicular to the stream with a small overhang off the edge of the hillslope in Star Creek and North York Creek watersheds in the fall prior to snow accumulation. Samples were collected directly from snowmelt troughs and snow bridges with clearly visible melt. Snowmelt was sampled, instead of the snowpack, to better reflect the meltwater signature during the snowmelt period (Johannessen and Henriksen, 1978; Williams et al., 2009). The timing of sample collection was based on access to backcountry sites and were taken opportunistically when crews were in the area and were able to observe active snowmelt.

**3.2.2 Soil water**

Suction lysimeters were installed between 30-60 cm depth using a hand auger in two locations near the toe of the hillslope in each sub-watershed in early spring 2014 (2015 for SEU; Figure 1). Suction lysimeters consisted of a 0.5 Bar ceramic cup and 38.1 mm PVC pipe to ensure ample water was collected for chemical analyses. Water from the suction lysimeter was sampled using a hand pump every two weeks between April and October in 2014 and 2015. Suction lysimeters were pumped dry

following sampling and pressure was applied. Thus, soil water was composed of water that was able to pass through the ceramic cup over the two-week period until the lysimeter was at equilibrium pressure with the surrounding soil. Shallow depths were targeted with the intention to collect the unsaturated soil water above the saturated zone in the hillslope, which was sampled separately.

**3.2.3 Hillslope groundwater**

Hillslope wells were installed with a shovel or hand auger to depth of refusal or maximum auger depth (1.5 m) near the hydrometric gauging stations at SEL, SEU, SWL and SWU (Figure 1). A site was added at SEU at the end of the summer in 2014, whereas the other sites were established during summer 2013. Wells were installed in three locations at each site: riparian, toe slope, and hillslope positions to determine the full range in hillslope groundwater. Well depths ranged between 0.5 m (riparian wells) and 1.6 m. Wells were purged using a hand pump prior to sampling. Samples were collected

approximately every two weeks, as available, between April and October in 2014 and 2015. Samples from the upper hillslope wells were generally only obtained during the snowmelt or high flow period; these wells were often dry during late summer. Riparian and toe slope wells contained water for all or most of the year, respectively. Water table depths were monitored with capacitance loggers (Odyssey, Dataflow Systems Ltd., New Zealand) at 10-minute intervals to identify the timing of shallow groundwater table responses to infer potential periods when hillslope-stream connectivity occurred.

### 3.2.4 Groundwater seeps

At the onset of this research, lack of access to backcountry sites restricted the installation of deep bedrock or till groundwater wells in upper sub-watersheds. Rather, groundwater seeps were used to characterize the possible range in groundwater signatures (both bedrock and till groundwater) within Star Creek. Seeps are defined here as areas of visible water seeping from hillslopes proximal to the stream or from small wetland areas further from the stream that form small tributaries or rivulets
that flow into the stream. The east and west forks were initially surveyed from the confluence with the main stem to the stream origins in the alpine area in July 2013. 25 visible seeps were identified which ranged in duration and magnitude of their contributions to streamflow. Some seeps were only active during the snowmelt season and recession period, reflecting streamflow dynamics. Other seeps were relatively stable throughout the entire snow-free period or throughout the winter baseflow period. Samples were collected during three flow conditions: high flow (May/June), recession flow (mid-July), and
baseflow (early September prior to fall rains), in both 2014 and 2015. This sampling campaign required more resources than for other sources, as a result sampling was completed only three times a year during hydrologically important extreme flow conditions rather than every two weeks from April to October as for other sources. Water temperature and electrical conductivity were measured with a handheld multimeter (YSI85, YSI Inc./Xylem Inc., Yellow Spring, Ohio, USA) during sample collection to aid in differentiating between deep bedrock groundwater, till groundwater and hillslope groundwater.

### 3.2.5 Bedrock and till groundwater

Preliminary end-member mixing analysis showed that a water source was missing from those initially collected (above) highlighting the need to characterize deeper groundwater. Due to monetary and access limitations, a single borehole was drilled to 12 m depth (15.2 cm in diameter) in the topographic ridge between SEL and SWL (approx. 500 m upstream from gauging sites) in October 2015 (Figure 1). Two wells were installed in the borehole, one well in a water-baring formation in the bedrock
at 11 m depth, and a second well in the glacial till deposits at 4.5 m depth, to characterize the differences in bedrock and till groundwater chemistry. Both wells had screens that were 1.5 m in length. Sand was used to backfill the borehole around the screened section of the bedrock groundwater well and was capped with bentonite clay. Local material removed during drilling was used to backfill the borehole up to the till layer. The same method of back filling (sand, bentonite clay, local material) was used for the till groundwater well. Bedrock and till wells were sampled every two to four weeks from April to October in 2016
and 2017. Water in the till well was purged until dry prior to sampling. Water in the bedrock well was purged for 2-5 minutes

prior to sampling because the recharge rate was faster than the pump rate. Water table depth and temperature were measured continuously with pressure transducers (HOBO U20, Onset Computer Corp., Bourne, MA, USA) at 10-minute intervals.

High concentrations of $Na^+$, $Cl^-$, and $SO_4^{2-}$ in till groundwater (Figure 3) and large variability between years suggested that the till groundwater well was likely contaminated by the bentonite clay used to backfill and seal between layers (Remenda and
van der Kamp, 1997). Slow recharge rates (and therefore, low hydraulic conductivity) of glacial till prevented the removal of three pipe volumes when sampling and the corresponding low hydraulic conductivity resulted in little flushing of bentonite contaminants. Faster recharge rates (and therefore, higher hydraulic conductivity) of the bedrock groundwater would aid in better flushing of bentonite contaminants which would reduce the effects on bedrock groundwater chemistry (Remenda and van der Kamp, 1997). As a result, the till groundwater samples were not included in the analyses herein; however, water table
depths and water temperature dynamics could still be used to understand the differences between till and bedrock groundwater responses and their roles in runoff generation.

## 4 Data Processing

End-member mixing analysis (EMMA) was used to visualize multi-variate source water and stream chemistry by reducing the dimensionality of the data with principal component analysis (PCA; Christophersen and Hooper, 1992). In addition, there were
multiple subjective decisions required prior to EMMA, such as choosing tracers/ions and defining sources. Two methods were used to determine if tracers were appropriate to use in the analysis, bivariate plots and TVR. First, a matrix of bivariate plots of stream chemistry data (ion concentrations), used most commonly in geographical hydrograph separations, was used to determine if ions were conservative in nature (Hooper, 2003). A linear relationship between tracers can be interpreted as a sign of conservative relationships. Second, TVR, used most commonly in sediment source apportionment studies, was used to
determine if the difference in ion concentrations between groups was larger than the variation within a source group (Pulley et al., 2015). TVR was calculated using the following equation for each tracer and compared between each source group pair:

$$\frac{\frac{\tilde{x}_{max} - \tilde{x}_{min}}{\tilde{x}_{min}} \; x \; 100}{mean\;(CV_{source\;1},\; CV_{source\;2})}$$

where $\tilde{x}_{max}$ is the maximum median tracer concentration of either source group, $\tilde{x}_{min}$ is the minimum median tracer concentration of either source group, and *CV* is the coefficient of variation (Pulley et al., 2015; Pulley and Collins, 2018). TVR should be greater than 2 to be considered appropriate for use in mixing calculations (Pulley and Collins, 2018), although
depending on the dataset in question, a greater threshold may be adopted to make the tracer selection more stringent and to help reduce the numbers of tracers included in further data processing.

Box and whisker plots and LDA were used to remove the subjectivity of defining sources (Ali et al., 2010; Pulley and Collins, 2018). Box and whisker plots were used as a visual means of discriminating between sources. LDA was then used to determine
if the combined sources exhibited sufficiently robust statistical separation (Pulley and Collins, 2018). LDA optimizes separation between the centroid of group clusters by partitioning the variation across each tracer and weighing that variation

into two axes (reducing the dimensionality). Other statistical classification methods, such as hierarchical clustering or k-means clustering, were not appropriate because source categories were known a priori. The data were processed in R (R Core Team, 2014) using 'lda' function in the MASS package (Venables and Ripley, 2002) to reduce dimensionality and assess the separation visually and the klaR ('stepwise' function; Weihs et al., 2005) package to model the data and determine the ability to separate groups statistically. The 'stepwise' function models the data while removing individual tracers iteratively. The 'backwards' direction was used in an attempt to maintain the most tracers with the 'lda' method, and 'ability to separate' criterion.

After the sources were characterized, the stream water was processed using principal component analysis (PCA; 'prcomp' function in R; R Core Team, 2014) as a method of dimensionality reduction to create a two-dimensional (2D) mixing space (Christophersen and Hooper, 1992). Stream water was standardized (subtracting the mean and dividing by the standard deviation for each sampling point), for each tracer, to create equal variance between chemical components and used to create a correlation matrix. PCA was conducted on the correlation matrix to calculate eigenvectors and eigenvalues. Standardized stream water was then projected into the end member mixing-space by multiplying by eigenvectors. Ideally, two principal components (PCs) explained most of the variation in the data and were used to generate a 2D mixing space, which corresponds to three sources in EMMA (Hooper, 2003). Other studies have used the 'Rule of 1' to determine how many dimensions, and therefore sources, should be used to create the mixing space (Ali et al., 2010; Barthold et al., 2011). For this study, the mixing space was set to two dimensions for ease of visualization but used all appropriate sources as presented by Inamdar et al. (2013) to provide a full description of potential source contributions. Source water was then standardized using stream water means and standard deviations for each ion and projected into the 2D mixing space as defined by the stream water (Christophersen and Hooper, 1992; Hooper, 2003). Stream water sources should create an outer boundary or polygon around all stream water samples if all sources were correctly identified and adequately sampled.

## 5 Results

### 5.1 Tracer and source water group selection

Bivariate plots were created and TVR was calculated to determine which tracers were appropriate for use in EMMA. Pearson correlation coefficients were calculated between all stream bivariate plots for stream water at each sub-watershed (Figure 4). These showed that all tracers exhibited acceptable linear trends with at least one other tracer (Pearson's r > 0.5, $p < 0.05$) and were thereby likely conservative in nature. TVR for almost all tracers at all sites were below 2 with the exception of precipitation group comparisons, which suggested that the within-group variation exceeded the between-group variation for all subsurface sources. Greater variation within- compared to between-source groups violates assumption 3) for EMMA (source water does not change) and was considered unacceptable. As a result, rather than calculating mixing ratios or percent contribution of sources to stream water on the basis of an un-mixing routine in EMMA, trends in stream water distribution were described in relation to source water dynamics and runoff processes.

The a priori classification of water sources was rain, snowmelt, soil water, hillslope groundwater, till groundwater and bedrock groundwater; however, not all sites conformed to these categories. Box and whisker plots showed that the distribution of rain and snowmelt were too similar to be considered as separate groups. Although riparian water mixes with stream water and should be chemically different from hillslope water as a source, soil water, toe slope water, and upper hillslope water were grouped with riparian water for most sites because the distribution of these samples were too similar to be considered separate sources. The exception was SEL and SWU in which riparian water was considered as a separate source. Final source water groups are described below for each sub-watershed. LDA plots indicated that LD1 and LD2 explained 88.5 % and 11.5 %, 95.3 % and 4.7 %, 81.1 % and 15.5 %, and 77.6 % and 22.4 % of the variance of the centroids for SWL, SWU, SEL, and SEU sites, respectively. Stepwise analyses were also used in attempt to reduce the redundancy of the tracers and to ensure that samples were well separated; on this basis, 99.7 %, 91 %, 98.6 %, and 99.9 % of samples were well separated in SWL, SWU, SEL, and SEU, respectively. In all sites, all tracers were retained to maximize the ability to distinguish between the source groups. Overall, these results support the conclusion that there was good separation between source water groups as re-categorized for the individual sites.

At the outset of this research, groundwater seeps were sampled in lieu of bedrock and till groundwater wells to characterize the variability in the chemical signature of groundwater throughout Star Creek. Most ion concentrations of the groundwater seeps were not chemically distinct because they were generally similar to stream water or hillslope groundwater in the PCA analyses (data not shown). However, the water temperature of groundwater seeps from spring to fall revealed that some seeps were consistently cool while others had larger fluctuations in temperature. This suggests that some seeps were potentially groundwater-fed and others were fed by shallow subsurface water, respectively. For example, in SEL, the temperature of a groundwater seep ranged between 2.2-3.7 °C throughout the summer (Figure 5), which is indicative of a bedrock groundwater source because the temperature range was muted and was largely not influenced by radiative warming (Taniguchi, 1993). In SEU, the temperature of a groundwater seep ranged from 2.5-3.5 °C (Figure 5), also indicating a bedrock groundwater source. Temperatures in the till groundwater well ranged between 2.7-9.7 °C, displaying some radiative heating and cooling, whereas the bedrock groundwater ranged between 5.1-5.8 °C, displaying little radiative effects (Figure 5). The groundwater seeps mentioned above had low variability like bedrock groundwater but were cooler suggesting potentially deeper bedrock groundwater sources than in the well. Temperatures of some other groundwater seeps were more similar to bedrock groundwater although more variable, ranging from 3.6-5.4 °C (data not shown), while others were more similar to till groundwater, ranging from 4.8-7.1 °C (Figure 5). The corresponding specific conductivity measurements add further complexity to these patterns. The cool, temporally more stable seeps had low conductivity from April to September, which was not reflective of the specific conductivity in the bedrock groundwater well. Rather, the other seeps with greater variability in temperatures had high specific conductivity, more consistent with the bedrock groundwater wells (Figure 5). Unfortunately, the till well specific conductivity could not be used due to the contamination mentioned above so it was unclear if the till groundwater had similar specific conductivity.

### 5.2 Source water characterization

#### 5.2.1 Star West source water

Water sources for the Star West Lower sub-watershed were grouped as precipitation (rain and snow), hillslope groundwater
(soil water, riparian water, and toe slope water), and bedrock groundwater and plotted in PCA mixing space (Figure 6). PC1 was mainly driven by cations and PC2 was driven by anions (Table 1). Minimal variation in chemistry across all precipitation samples (standard deviation (SD) of 2.4 and 1.1 for PC1 and PC2, respectively) and overlap of snow and rain samples in the mixing space confirmed that it was appropriate to aggregate all samples (snow and rain) taken across all sites (Star Creek, York Creek, and Coleman). Hillslope groundwater exhibited greater chemical variation across samples (SD of 3.8 and 2.0 for
PC1 and PC2, respectively) compared to bedrock groundwater (SD of 2.9 and 4.8 for PC1 and PC2, respectively), but no clear temporal pattern was observed. Bedrock groundwater chemistry showed slight temporal variation with more positive values in PC2 in the spring than in the fall.

Water sources for the Star West Upper sub-watershed were similarly grouped as precipitation (rain and snow) and hillslope groundwater (soil water, toe slope water, and upper hillslope water), but here riparian water displayed greater difference from
330 hillslope groundwater and was considered as a separate source (Figure 6). Bedrock groundwater samples were collected from a lower elevation in the watershed and may not be representative of higher elevation groundwater chemical composition; therefore, they were excluded from the analysis for the upper sites. Further, there were only two seeps identified in the upper watershed, but the temperature and chemical composition of these seeps were not reflective of bedrock groundwater. While this did not exclude bedrock groundwater contributions to streamflow in the upper regions of the watershed, it showed the
335 chemical composition of the bedrock well and the two seeps may not have been representative of the bedrock groundwater chemistry in the Star West Upper sub-watershed. Precipitation clustered tightly in one location except for four snow samples and one rain sample, which increased the SD for precipitation (SD of 2.7 and 2.3 for PC1 and PC2, respectively). All sources showed similar variation as precipitation; hillslope groundwater had a SD of 4.3 and 2.7 for PC1 and PC2, respectively and riparian water had a SD of 3.0 and 2.0 for PC1 and PC2, respectively. A temporal pattern was observed for hillslope water in
which hillslope water became less like precipitation from spring to fall (Figure 7). Temporal variation was also observed across months for riparian water, in which $SO_4^{2-}$ concentrations increased from spring to fall (Figure 7).

#### 5.2.2 Star East source water

Water sources for the Star East Lower sub-watershed were grouped as precipitation (snow and rain), soil water, riparian water, groundwater seep and bedrock groundwater (Figure 8). Precipitation (SD of 1.4 and 1.1 for PC1 and PC2, respectively) and
345 bedrock sources (SD of 1.0 and 0.9 for PC1 and PC2, respectively) were the same as those used in SWL. Hillslope groundwater samples were initially grouped together as a single source but high standard deviations and clustering within the group suggested the separation of riparian water (SD of 1.0 and 1.5 for PC1 and PC2, respectively) and soil water (SD of 4.3 and 1.5 for PC1 and PC2, respectively) as individual sources. Soil water was most different from stream water and varied from spring

to fall (increased $Ca^{2+}$ and $Mg^{2+}$ concentrations; Figure 9). Riparian water was most similar to stream water and did not vary over the season. A single groundwater seep that was chemically similar to stream water but temperatures were consistently cool was retained to aid in the explanation of stream water dynamics (Figure 8).

Water sources for the Star East Upper sub-watershed were grouped as precipitation (rain and snow), hillslope groundwater (soil water, riparian water, and toe slope water), and groundwater seep (Figure 8). Precipitation displayed little variation (SD of 2.4 and 1.1 for PC1 and PC2, respectively). Large variation was observed for hillslope groundwater (SD of 9.3 and 7.3 for PC1 and PC2, respectively). Toe slope water and riparian water had some chemical dissimilarities but were not different enough from each other, or soil water, to be considered as different groups. Some temporal variability was observed in riparian water compared to Star East Lower; however, soil water had much larger temporal variability than riparian water (Figure 9). A single groundwater seep was identified in SEU. The seep was chemically similar to stream water but temperatures were consistently cool and indicative of a deep groundwater source so it was retained to aid in the explanation of stream water dynamics (Figures 8 and 9).

### 5.3 Stream water characterization

Stream water chemistry for all four sites showed high temporal variation throughout the months of open-water flow (Apr-Oct) but little variation between years. As a result, the temporal pattern of stream water was characterized for each site in general for both 2014 and 2015 combined. Further, due to the lack of source water samples during winter months, the temporal pattern of stream water was characterized from April to October, which represents the most dynamic hydrologic period from the beginning of snowmelt through to the start of the next year's snow accumulation period. Hydrologic characteristics of the 2014 and 2015 water years are indicated in Table 2.

### 5.3.1 Star West stream water

The first two principal components (PCs) from the PCA analysis explained 87 % and 77 % of the variation in stream water chemistry in Star West Lower and Star West Upper streams, respectively. Temporal variation of stream water chemistry was constrained within the broader multi-variate mixing space created by the variation in source water chemistry, but not within the more constrained mixing space of the mean composition (+/- 1 std. dev.) of these sources (Figure 6). In April, stream water was most similar to the hillslope groundwater (and riparian water in SWU). Stream water transitioned through May to become most similar to precipitation source water in June (and July in SWU). In SWL, stream water was slightly more similar to hillslope groundwater and bedrock groundwater in July. In August, September and October, stream water chemistry was more variable and was similar to precipitation and hillslope and bedrock groundwater. The temporal pattern associated with variation in stream water chemistry through the fall was perpendicular to the direction of the bedrock temporal pattern, suggesting that hillslope groundwater (soil water, toe slope water and riparian water), rather than bedrock groundwater, was driving variation in stream water chemistry in the fall in SWL. Time series of $Ca^{2+}$ and $SO_4^{2-}$ show that hillslope groundwater was most similar to stream water (Figure 10). Stream water in SWU was again more chemically similar to hillslope groundwater and riparian

water through August, September, and October, but stream water chemistry differed slightly from its chemical composition in the early summer months. Riparian water chemistry had a similar temporal shift from April to October as stream water chemistry whereas hillslope groundwater and soil water had greater temporal variation (Figure 7). Further, water table depth in the hillslope well indicates that the upper hillslope is largely disconnected from the stream in the fall in both SWL and SWU (Figure 11) so it is more likely that the riparian area is contributing flow to the stream in the fall.

### 5.3.2 Star East stream water

Temporal patterns of variation in stream water chemistry observed for Star East Lower and Star East Upper were very consistent with each other, again with the exception of the bedrock groundwater well which was only sampled at a lower elevation site and therefore not included in the PCA analysis in Star East Upper. However, seeps that displayed temporal stability in water temperature typically characteristic of deep groundwater (Figure 5) were used in the analysis for both SEL and SEU. The first two PCs explained 86 % and 83 % of the variation in stream water chemistry in Star East Lower and Star East Upper, respectively (Figure 8). For both sub-watersheds, temporal variation in stream water chemistry were mostly constrained within the mixing space produced by the variation in source water chemistry, except during September/October when stream water plotted outside this boundary. In April, stream water was most similar to the riparian/hillslope water (or bedrock groundwater for Star East Lower). The chemistry of stream water transitioned through May and was most similar to precipitation in June. In July and August, stream water became dissimilar from precipitation and was once again similar to riparian/hillslope water or bedrock groundwater. In September and October, stream water was less similar to riparian/hillslope water and plotted outside the mixing space of the identified sources. Since stream water was not contained within the boundary created by the source water, it is likely that an additional source was not captured by field sampling. However, the temporal variation in the chemistry of the groundwater seep followed the same pattern as the September/October stream water in both sub-watersheds, suggesting the same source water for the groundwater seep and late fall baseflow (Figure 8). Consistently cool temperatures of the seep in Star East Lower (2.2-3.7 °C) and Star East Upper (2.5-3.5 °C) suggest a deeper groundwater source.

### 6 Discussion

Twice monthly stream water and source water samples collected in Star Creek from April to October in 2014 and 2015 have been used here to conceptualize runoff generation in Alberta's Rocky Mountains. Results from this study allow for detailed examination of temporal patterns in source water chemistry and a qualitative description of source water contributions to stream water. While our intention was a quantitative estimate of source water contributions to streamflow using an un-mixing routine, two of the key assumptions for EMMA, the chemical composition of sources does not change over 1) the time scale considered or 2) with space (Hooper, 2003; Inamdar, 2011), were violated in this dataset. Source water chemistry varied greatly across the watersheds. For example, when all hillslope samples from each sub-watershed were projected into the mixing space created by stream water at the watershed outlet (SM), large variability was evident between sites (Figure 12). While there was

some overlap between some sites (SWL and SEU), SWU was clearly different than the other hillslope samples. As a result, source water from within individual sub-watersheds was used to reduce the uncertainty associated with large spatial variability in source water chemistry. However, the variability within sites was also quite large. The CV of source water tracer

concentrations was often larger than the CV of the stream water tracer concentrations (there should be little to no variation in source water over time; James and Roulet, 2006; Inamdar, 2011), particularly for $K^+$. The occasions where source water CV were smaller than stream water CV for most ions were for seeps in SEU and SEL, bedrock groundwater in SWL and SEL, and hillslope and riparian water in SWU. Chemical signatures of source water have been shown to vary seasonally and annually (Rademacher et al., 2005) as well as spatially across sub-watersheds in southern Quebec, Canada (James and Roulet, 2006).

As a result, James and Roulet (2006) suggested that only source water from within individual sub-watersheds of interest should be used in un-mixing calculations. Inamdar et al. (2013) further argued that mixing proportions should not be calculated because multiple assumptions are often violated and can lead to significant errors in un-mixing proportions. Rather, temporal and spatial variation in stream water and source water should be examined and used to describe or to develop a physically-based conceptualization of runoff mechanisms.

The inability to run the unmixing routine (stream water fell outside the bounds of the source water) also hindered the use of some tracer selection methods. Other studies have often used the selection criteria presented in Barthold et al. (2011) but the unmixing routine is required for this method. Rather, the TVR and LDA have been presented as effective parameters to subjectively determine if tracers are included in the analysis and if sources are well separated or grouped appropriately, respectively (Pulley et al., 2015; Pulley and Collins, 2018; and others – see comprehensive review in Collins et al., 2017).

Despite the violation of assumptions, notable temporal trends in source water chemistry were observed in snowmelt, riparian water, hillslope and soil water, and bedrock groundwater and their contributions to stream water can be generalized for all sub-watersheds in Star Creek in a number of ways. The water that was stored in the hillslope overwinter (or "reacted water") was likely the first to reach the stream in the early spring prior to high flow as snowmelt started to saturate the landscape (Figures 6 and 7). Temporal patterns in stream water chemistry also showed a spike in concentrations of some ions (e.g., $Ca^{2+}$

concentration in Figure 11) in the stream in the early spring as this reacted water mobilized prior to the onset of the snowmelt freshet. Although three snowmelt samples in 2014 showed similar ionic pulses early in the snowmelt season to those reported in the Colorado Rocky Mountains (Williams et al., 2009), the concentrations were notably less than from all other sources and thus not likely an important source of the observed early season increase in stream water concentration of some ions. Rather, the delivery of reacted water to the stream at the onset of snowmelt is likely similar to the flushing mechanism observed in the

Turkey Lakes Watershed in central Ontario, Canada (Creed and Band, 1998) where high nitrogen concentrations were observed prior to peak streamflow. McGlynn et al. (1999) observed the displacement of old water to the stream at the onset of snowmelt in Sleepers River Research Watershed in Vermont, USA and suggested this was due to a small volume of snowmelt being added to a large storage of water already in the subsurface.

This initial displacement of reacted water was followed by a dilution effect, where large volumes of low concentration

snowmelt mixed with soil water and contributed to streamflow. Snowmelt was the major event that produced a water table

response in all hillslope wells and connected the hillslopes to the stream (Figure 11; Spencer et al., 2019). The initial snowmelt period was also the only time overland flow was observed at the study site. Other studies have also reported that snowmelt creates a dilution response in the stream (Rademacher et al., 2005; Cowie et al., 2017). Conversely, the opposite has been observed whereby a previously disconnected source was connected to the stream and caused an increase in solute

concentrations (McNamara et al., 2005). Although this was the main period of hydrologic connectivity in Star Creek (Figure 11), we did not observe an increase in stream water ion concentrations associated with newly connected sources. Hillslope groundwater and soil water chemistry should reflect the dilution from snowmelt and the subsequent drying of the hillslope, thereby increasing ion concentrations in the soil water from spring to fall. This corresponding temporal pattern in hillslope groundwater chemistry was observed in SWU (Figure 7) and for soil water chemistry in SEL and SEU (Figure 9).

Source water contributions to the stream were more similar within Star East (SEL and SEU) and Star West (SWL and SWU) sub-watersheds than between alpine/sub-alpine (SEU and SWU) and upper montane (SEL and SWL) sub-watersheds. PCA plots for SEL and SEU showed that stream water chemistry was most similar to precipitation in May and June, whereas a dilution effect from snowmelt occurred in June and July in SWL and May to July in SWU. It should be noted that although dilution was observed, stream water was less dilute than snowmelt alone. Snowmelt water interacts with the soil as it moves

through the subsurface to the stream directly influencing the chemical composition of the snowmelt contributions to streams (Sueker et al., 2000). The delayed response in SWL and SWU is consistent with the watershed storage estimates from baseflow recession analysis (Spencer et al., 2019) that suggested that the west fork sub-watersheds had a larger storage capacity than the east fork sub-watersheds. Accordingly, more water would be required to fill storage before saturation or hydrologic connectivity occurred.

Differences in the east and west forks were also evident in the hysteresis pattern in stream water chemistry from spring to fall. Star West sub-watersheds had a counterclockwise pattern, whereas Star East sub-watersheds had a clockwise pattern. In general, this is an artefact of the PCA analysis driven by the specific ions that defined each PC (Table 1). In Star East, the first PCs were dominated by anions and the second PCs were dominated by $SO_4^{2-}$ (negative relationship). While the first PCs for Star West were dominated by anions, the second PCs included a mix of anions and cations and $SO_4^{2-}$ with a positive correlation

thereby producing an opposite hysteresis pattern. Although this is an artefact of the PCA analysis, it was ultimately due to slight variations in the sources contributing to the streams at different times during the flow season. For instance, in SWL and SWU, stream water was chemically similar to riparian water in the fall (Figure 6); whereas in SEL and SEU, stream water was similar to hillslope groundwater in August but fell outside the boundaries created by the identified sources in September and October (Figure 7). Details on the possible processes underlying these differences are described below.

Temporal variations in riparian water in SWU were observed from spring to fall and followed the same pattern observed in stream water chemistry in May compared to September/October (Figure 7). It is not clear if the stream chemistry responded to variation in riparian chemistry or if riparian water responded to stream chemistry, but these pools of water were likely mixing to create similar temporal patterns rather than reflecting those of hillslope water chemistry influencing the stream. Timing of riparian and stream water level responses may be used to help clarify these patterns in future research. Other studies

have shown the importance of the riparian zone for "buffering" stream water chemistry from inputs from other sources particularly for individual hydrologic events (McGlynn and Seibert, 2003; Grabs et al., 2012). Further research needs to be conducted to estimate the extent of the riparian area and the potential volume of water that may contribute to streamflow in Star East compared to Star West.

A groundwater seep in SEL and SEU followed similar temporal patterns as stream water from spring to fall (Figure 7) and
may provide insights into the sources of stream water in September and October. Consistently cool, but low specific conductivity of the groundwater seeps suggest they likely reflected a deeper bedrock groundwater source different than the bedrock groundwater well. Although most of the stream is situated within the same geologic formation, there may be differences in bedrock groundwater chemistry associated with heterogeneous sedimentary layers or contact time in the upper versus lower watershed (Freeze and Cherry, 1979). Temperature signals from other seeps suggested some were fed by shallow
subsurface water or till groundwater (larger fluctuations in temperature; Figure 5; Taniguchi, 1993), yet they had high specific conductivity and similar ion concentrations as bedrock and hillslope groundwater. This suggests that there are likely many complex subsurface flow pathways making it difficult to differentiate between subsurface sources, but it is possible, therefore, that additional bedrock groundwater sources were contributing to streamflow in Star East in September and October. Other tracers such as oxygen and hydrogen isotopes or non-conservative tracers such as nitrogen and dissolved organic carbon may
help to better differentiate between seeps, hillslope groundwater, and bedrock groundwater (e.g., Cowie et al., 2017; Ali et al., 2010; Orlova and Branfireun, 2014). Additional observation wells in the bedrock and till would be required to characterize more thoroughly the variability in groundwater across the watershed (Rinderer et al., 2014).

Topographic transitions and convergent zones have been associated with groundwater contributions to streamflow (Covino and McGlynn, 2007; Hjerdt et al., 2004). While minimal groundwater discharge occurred over the mountain front recharge
zone in Humphrey Creek, southwest Montana, USA, considerable groundwater discharge was observed in the valley bottoms (Covino and McGlynn, 2007). Large increases in the concentration of stream water ions that may be associated with strong groundwater upwelling were not evident between April and October or along the length of the stream. However, chemical signatures of groundwater seeps suggest that some bedrock groundwater may not be distinguishable from stream water. Consequently, these transitions may not be visible in water chemistry along the length of the stream.

Although contamination of the till groundwater well limited the inferences that could be made from its water chemistry, water levels in the bedrock and till groundwater wells do provide some insights into potential contributions of till groundwater to streamflow. Water table depth in the till groundwater was more responsive than bedrock groundwater level in the spring, though the overall rise in water level in the bedrock was slightly greater. Despite the flashier response earlier in the year, till groundwater levels remained elevated longer than bedrock groundwater resulting in a slower recession (slower drainage) in
the till groundwater well in the summer (Figure 13). Similar to the post-glacial landscape in Sleepers River watershed (Shanley et al. 2015), slower drainage from till groundwater may be partly responsible for maintaining streamflow during late summer or fall. The temperature range of some seeps sampled along the stream length were similar to till groundwater, but ion concentrations were similar to hillslope and bedrock groundwater. It is likely that glacial till chemistry is similar to hillslope

and bedrock groundwater given that they are situated above and below the glacial till layers. Heterogeneous glacial till deposits with different physical characteristics were also linked to the variable release of stored water, and thus the variability in baseflow, in the Scottish Highlands (Blumstock et al., 2015). Glacial till in the Rocky Mountains can be highly spatially variable likely promoting multiple flow pathways (Langston et al., 2011). Clay lenses can create perched water tables that have different response times than the rest of the till matrix (Evans et al., 2000) or create complex groundwater flow pathways (Freeze and Witherspoon, 1967). Further research is needed to help differentiate between bedrock groundwater and till groundwater and their contribution to stream water during low flows.

**7 Conclusions**

Stream and source water were collected over the 2014 and 2015 water years and visualized using principal components analysis to conceptualize runoff generation processes in the Canadian Rocky Mountains. While strong variability in source water chemistry limited our ability to quantitatively estimate the relative contributions of multiple water sources to the stream using an un-mixing routine, the analyses used here enabled a strong qualitative description of precipitation, hillslope water and bedrock groundwater source contributions to streamflow. This allowed us to both indirectly observe and infer key runoff generation processes in watersheds with a complex lithological structure characteristic of the highly permeable bedrock and glacial till of the Alberta Rocky Mountain region.

Stream water chemistry in four sub-watersheds of Star Creek showed that Star East (SEL and SEU) and Star West (SWL and SWU) sub-watersheds were more similar than alpine/sub-alpine (SEU and SWU) and upper-montane (SEL and SWL) physiographic zones. In general, higher concentration "reacted" water reached the stream first at the onset of spring melt in all sub-watersheds. This was followed by a dilution effect as the snowmelt saturated the landscape and the hillslope was connected to the stream. Fall baseflows differed between Star East and Star West forks. Star West stream water was once again similar to hillslope water or riparian water, but Star East stream water plotted outside the boundary of the measured sources. Seep water temperatures were cool and had low variability suggesting it may be deeper bedrock water contributing to the stream. Slower recession rates (and likely lower hydraulic conductivity) in the till groundwater well than in the bedrock groundwater well suggest that water recharged into the till groundwater may be slowly released to the stream. Contamination of the till groundwater well made it unclear when it was contributing to the stream, but groundwater table fluctuations suggested it is likely contributing during late summer or fall. More research on the variability of bedrock and till groundwater chemistry is needed to clarify the difference between these sources and their contributions to streamflow throughout the year. However, it is clear from this research that multiple subsurface flow systems lead to the slow leakage of bedrock and till groundwater to the stream promoting higher baseflows in this region compared to regions with shallow soils and impermeable bedrock where groundwater stops flowing in the summer.

## Author contributions

US and AEA secured funding enabling this research and supervised the research project. SS designed and carried out the field research. SS analysed and interpreted the data with assistance from all co-authors, particularly ALC with statistics and R code. SS prepared the original draft of the manuscript with contributions in review and editing from all co-authors.

## Data availability

Please contact the corresponding author for data availability.

## Competing interests

The authors declare no conflicts of interest.

## Acknowledgements

The authors would like to acknowledge funding support from Alberta Agriculture and Forestry, Alberta Innovates Water Innovation Program, Forest Resource Improvement Association of Alberta (FRIAA), Canadian Forest Products Ltd., and the 555 National Science and Engineering Research Council (NSERC). Rothamsted Research receives strategic funding from UKRI-BBSRC (UK Research and Innovation-Biotechnology and Biological Sciences Research Council) and the contribution to this paper by ALC was supported by grant BBS/E/C/000I0330. This work would not have been possible without help in the field by Chris Williams, Amanda Martens, Melaina Weiss, Evan Esch, Veronica Martens, Eric Lastiwka, Kirk Hawthorn, Kalli Herlein, Shauna Stack, Chrystyn Skinner, Aryn Sherrit, Erin Cherlet, and Mike Pekrul. We would like to thank Kevin Devito 560 for his contribution to ideas during the initial stages of this research. We would also like to thank three anonymous reviews for their comments and suggestions which have substantially improved this manuscript.

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

**Tables**

**Table 1. Ions that explained the most variation in PC1 and PC2 for each sub-watershed in Star Creek.**

| | PC1 | PC2 | | PC1 | PC2 |
|---|---|---|---|---|---|
| **SEL** | Mg (-) | SO$_4$ (-) | **SWL** | Na (-) | K (-) |
| | Si (-) | Cl (+) | | Mg (-) | SO$_4$ (+) |
| | Ca (-) | | | Ca (-) | Cl (+) |
| | Na (-) | | | Si (-) | |
| | K (-) | | | | |
| **SEU** | Mg (-) | SO$_4$ (-) | **SWU** | Mg (-) | Cl (+) |
| | Ca (-) | Si (+) | | Na (-) | Ca (-) |
| | Na (-) | | | Si (-) | K (-) |
| | K (-) | | | | SO$_4$ (+) |

**Table 2. Streamflow and precipitation metrics for 2014 and 2015 water years.**

| | 2014 | | 2015 | |
|---|---|---|---|---|
| | SW | SE | SW | SE |
| Annual precipitation (mm) | 1149 | 1089 | 1091 | 1090 |
| Annual discharge (mm) | 944 | 648 | 719 | 468 |
| Proportion of discharge May-July | 0.69 | 0.74 | 0.45 | 0.54 |
| Peak discharge (m$^3$/s) | 1.20 | 0.75 | 1.14 | 0.72 |
| Average daily discharge* (m$^3$/s) | 0.14 (± 0.20) | 0.08 (± 0.12) | 0.10 (± 0.10) | 0.06 (± 0.07) |

*standard deviation in brackets

**Figure Captions**

Figure 1: Star Creek watershed. Suction lysimeter and hillslope groundwater well locations are magnified in green boxes.

Figure 2: Star East (left) and Star West (right) sub-watersheds. The Star East alpine area is more constrained and smaller than the Star West alpine area. Both sub-watersheds have steep headwalls with talus slopes in the alpine zone.

Figure 3: Box plots for Star West Upper (SWU), Star East Upper (SEU), Star West Lower (SWL), and Star East Lower (SEL) showing the ranges in chemistry for potential sources.

Figure 4: Bivariate plots of stream water chemistry at a) Star East Lower, b) Star East Upper, c) Star West Lower and d) Star West Upper. Top half of plots represents the Pearson's correlation coefficient (r) for the linear relationship between each solute.

Figure 5. Box and whisker plot of groundwater, seep, and stream water temperature (left) and specific conductivity (right). Solid line indicates the median and the dashed line indicated the mean. The box indicates the $25^{th}$ and $75^{th}$ percentiles, the whiskers indicate the $10^{th}$ and $90^{th}$ percentiles, and the circles indicate points within the $5^{th}$ and $95^{th}$ percentiles. "Other seeps" is shown here as an example of the temperature and specific conductivity in many of the other seeps that were identified in the watershed but not used in the PCA biplots.

Figure 6: The first two principal components (PC) of variation in stream water chemistry in Star West Lower (left) and Star West Upper (right) from April to October (values in brackets indicate the percent variation explained by each PC). Square symbols indicate the mean chemical composition (+/- 1 standard deviation) of stream water sources for each sub-watershed.

Figure 7: Time series of Si and $SO_4^{2-}$ concentration in Star West Upper stream and source water in 2014 and 2015.

Figure 8: The first two principal components of variation in stream water chemistry in Star East Lower (left) and Star East Upper (right) from April to October (values in brackets indicate the percent variation explained by each PC). Square symbols indicate the mean chemical composition (+/- 1 standard deviation) of stream water sources for each sub-watershed.

Figure 9: Time series of $Mg^{2+}$ concentration for Star East Lower (top) and Star East Upper (bottom) stream and source water in 2014 and 2015.

Figure 10: Time series of $Ca^{2+}$ and $SO_4^{2-}$ concentration in Star West Lower stream and source water in 2014 and 2015.

Figure 11: Observed inputs (snow depth and daily precipitation – estimated snow and rain proportions) and responses (stream discharge, stream Ca2+ concentration, shallow groundwater wells – hillslope and riparian) for Star Creek sub-watersheds in 2014 (left) and 2015 (right). Precipitation phase was separated into snow and rain after Kienzle (2008).

Figure 12: Hillslope groundwater from all sub-watershed sites in 2-D mixing space, which was derived from principal component analysis of Star Main (Figure 1) stream water. PC1 and PC2 represent the first and second principal components

Figure 13: Bedrock groundwater and till groundwater well responses (depth below ground, m) over the 2017 calendar year.