# Peer review of "Hillslope and groundwater contributions to streamflow in a Rocky Mountain watershed underlain by glacial till and fractured sedimentary bedrock"

_Hydrology and Earth System Sciences, 2020_

## Referee Comment (RC1) · Anonymous Referee #1 · 13 Apr 2020

Review of the manuscript 'Seasonally varied hillslope and groundwater contributions to streamflow in a glacial till and fractured sedimentary bedrock dominated Rocky Mountain watershed' by Spencer et al.

General comment This works uses hydrochemical data to describe and infer runoff generation processes in the subcatchments of the Rocky Mountains. The topic is certainly interesting for the readership of HESS. The manuscript is generally well written. However, there are two main points that do not sound convincing to me: i) the focus on catchment resilience and disturbance, that do not appear to be logically linked to the

investigations carried out and sounds out of context; ii) the presence of hydrochemical data only: despite the powerful nature of hydrochemistry as hydrological tracer, the combination of racer data and hydrometric data can help to unravel the complexity of hydrological processes at the catchment scales. Thus, the manuscript fails to describe in a robust, quantitative, and convincing way how water moves through this landscape in response to both rainfall and snowmelt. As a result, a clear contribution of this study to the body of knowledge is not evident. Please, find some specific and minor comments below.

Specific comments - The abstract is a bit vague. The motivation sounds weak, there are no specific objectives, the methods are partly unclear (water sources were sampled for what kind of analysis?), and the concept of hydrological resilience is not specified. I suggest revising it entirely.

- The Introduction fails to clearly stress what it is not well known about the specific topic and what is the main research gap, and the reader, at the end of the Introduction is left wondering why another study on streamflow contribution is needed. An overall objective and testable hypothesis is not reported. The two specific objectives are introduced quite abruptly, without a clear and logical connection with the paragraph above. I suggest to heavily revise the Introduction to keep these points into consideration.

- 190-208. I suggest to consider the work by Barthold (2001) and to specify the reported approaches were preferred over this method. Moreover, briefly mention how TVR and LDA work to allow the reader better understanding the methods that were used. https://agupubs.onlinelibrary.wiley.com/doi/pdf/10.1029/2011WR010604

- I suggest merging Figs. 5 and 6 (making a multi-panel figure) and sections 5.2.1 ad 5.2.2, and Fig. 7 and 8 and sections 5.2.3 and 5.2.4 in order to present the results from the two subcatchments more organically. Similarly, I recommend merging Section 5.3.1 and 5.3.2 (Star West), and 5.3.3 and 5.3.4 (Star East) to avoid too much text and results in fragmentation.

- Since there is, at least in some cases, a strong seasonal pattern in hydrochemistry, I suggest considering making a time series plot of the different water sources in the two subcatchments in order to show, for instance, when and to which extent the stream water signature gets closer to that of hillslope groundwater and riparian water. In addition, the Authors might consider adding times series of groundwater temperatures or boxplots, as this tracer is part of the story and was shown to be able to partly explain groundwater contributions to streamflow.

- 417-418. Which evidence do the Authors have to infer the temporal dynamics of hillslope water moving to the stream? Moreover, how could the Authors describe old water mobilization without having quantified its proportion in stream water? Or this is a general statement not based on the presented dataset? Please, explain.

- 464-474. I feel this part is quite out-of-context and disconnected from the previous discussion. In general, I think that focusing on catchment resilience is not so straightforward and sound a bit contrived to me. The same comment applies to the Conclusions.

Minor comments and technical corrections 1. The title is long and complex. I suggest making it more compact and clearer.

11. I suggest to change as follows: "A lack of . . .but mechanisms governing. . .".

13. ". . .although much. . .": I cannot see the logical link in this sentence. Please revise.

13-14. "to interpret how forest disturbance may impact streamflow quantity". I would not focus on understanding runoff generation processes to this aim, but mostly on the ecohydrological role of forest on streamflow. Please, revise.

22. "but was unlike the measured sources": this sentence is not clear before reading the abstract. Please, clarify.

29: Perhaps put it more general, mentioning pathogens.

35. What do the Authors refer to by "features"? Please explain.

[Figure]

112. "...a priori...": Was there any evidence, field observation, previous study or knowledge of the area that allowed for this assumption?

193. TVR: please report the definition and possibly the equation to let the reader immediately understand it.

229-230. This sentence is not clear to me (without reading the cited references). Please specify.

245. leu?

269: Perhaps add "compared to bedrock groundwater".

Fig. 6b). Could the Authors perhaps colour-code samples for season (spring, summer, fall)?

322. Which are these months?

340. Why a source might be missing? Please, explain.

393-394: Are groundwater levels available? Their temporal patterns could help understand which feeds which. Perhaps some piezometers could be installed for a follow-up study.

429-430. What does "increase in stream water chemistry" mean? Moreover, how would be possible to infer connectivity through hydrochemical data only? Some speculations could be done but a combination of hydrometric and tracer data would serve this purpose better.

431. Contributions to what? Please specify.

433. It cannot be all rain water, can it? Please, revise/explain.

Possible useful readings for additional analyses and for the discussions section: Correa, A., Breuer, L., Crespo, P., Célleri, R., Feyen, J., Birkel, C., Silva, C., Windhorst, D., 2019. Spatially distributed hydro-chemical data with temporally high-resolution is needed to adequately assess the hydrological functioning of headwater catchments. Science of The Total Environment 651, 1613–1626. https://doi.org/10.1016/j.scitotenv.2018.09.189

Godsey, S.E., Hartmann, J., Kirchner, J.W., 2019. Catchment chemostasis revisited: Water quality responds differently to variations in weather and climate. Hydrological Processes 33, 3056–3069. https://doi.org/10.1002/hyp.13554

Hoeg, S., Uhlenbrook, S. and Leibundgut, C., 2000. Hydrograph separation in a mountainous catchment — combining hydrochemical and isotopic tracers. Hydrol. Process., 14: 1199-1216. doi:10.1002/(SICI)1099-1085(200005)14:7<1199::AID-HYP35>3.0.CO;2-K

Hrachowitz, M., Bohte, R., Mul, M.L., Bogaard, T.A., Savenije, H.H.G., Uhlenbrook, S., 2011. On the value of combined event runoff and tracer analysis to improve understanding of catchment functioning in a data-scarce semi-arid area. Hydrol. Earth Syst. Sci. 15, 2007–2024. https://doi.org/10.5194/hess-15-2007-2011 Hydrograph separation in a mountainous catchment - combining hydrochemical and isotopic tracers, 1999. 18.

Nadal-Romero, E., Khorchani, M., Lasanta, T., García-Ruiz, J.M., 2019. Runoff and Solute Outputs under Different Land Uses: Long-Term Results from a Mediterranean Mountain Experimental Station. Water 11, 976. https://doi.org/10.3390/w11050976

Penna, D., van Meerveld, H.J., Zuecco, G., Dalla Fontana, G., Borga, M., 2016. Hydrological response of an Alpine catchment to rainfall and snowmelt events. Journal of Hydrology 537, 382–397. https://doi.org/10.1016/j.jhydrol.2016.03.040

Suecker, J.K., Ryan, J.N., Kendall, C., Jarrett, R.D., 2000. Determination of hydrologic pathways during snowmelt for alpine/subalpine basins, Rocky Mountain National Park, Colorado. Water Resour. Res. 36, 63–75. https://doi.org/10.1029/1999WR900296

105, 2020.
Interactive
comment

---

## Referee Comment (RC2) · Anonymous Referee #2 · 5 May 2020

Reviewer comments for:

Seasonally varied hillslope and groundwater contributions to streamflow in a glacial till and fractured sedimentary bedrock dominated Rocky Mountain watershed.
Spencer et al.
In Hydrology and Earth System Sciences

Overview

This is an interesting research manuscript with well-defined objectives of using EMMA to understand relative variation in stream water sources in forested and alpine watersheds of the Canadian Rocky Mountains. Overall the manuscript is sound and has relatively minor grammatical errors. However, it is suggested that the introduction and discussion sections more clearly identify the secondary objectives and how the study . It is unclear if the results of this study have a primary goal of improving understanding of the variability between different groundwater sources of (e.g. bedrock groundwater vs. glacial till groundwater) in alpine and sub-alpine watersheds, or to understand/predict how future forest disturbance will impact watershed hydrology and runoff generation.

It is suggested that the study site description be expanded to more clearly quantify the size and extent of surface and sub-surface biological and geological features (talus slopes, alpine areas, glacial till, forested areas, riparian areas, etc.).  It was hard to determine where and why the division of upper and lower sub-watersheds was chosen for this study and how that division was critical to addressing the question of the impacts of forest disturbance on hydrologic disturbance. Both upper and lower sub-watersheds had forested areas so the separation of upper and lower sub-watersheds did not appear to be a proxy for forested vs unforested/disturbed areas.

Secondly, the discussion and conclusion sections should be expanded to more clearly indicate how the results in this study advance or improve the scientific understanding of "how forest disturbance may impact streamflow quantity". It is unclear if the study was able to confirm or reject the hypothesis stated in the abstract that "slow release of groundwater from glacial till" (line 24) generates "hydrologic resilience" in the Rocky Mountains?

Line 18:
Suggest defining "old water" as related to time the water has spent in the watershed rather than the true age of water.

Line 20:
In Star east in September and October the stream water was unlike the sources. What is the additional source or is it a mixed signal?

Same statement is in the conclusion but the proposed explanation of the "missing" sources is either absent or not clear in the conclusion. Clarification would be helpful.

Line 29:
 What is the specific reference for beetle infestation?  A few Studies from the Rocky Mountains to consider reviewing,

Pugh & Small, 2011.
**https://doi.org/10.1002/eco.239**

Bearup *et al.*, 2014.
https://doi.org/10.1038/nclimate2198

Line 53:
Consider reference of Cowie *et al*. (2017) here as that study does use EMMA to examine potential source waters from bedrock groundwater, glacial till groundwater, talus slope water, and soil water on streamflow contributions in forested and alpine watersheds in the Rocky Mountains.

Line 77:
Define area weighted precipitation. Was precipitation measured at multiple elevations? With > 1000m elevation change how much does the total precipitation change over that gradient? One suggestion is use of a hypsometric curve to distribute precipitation over elevation (see Cowie et al., 2017)

Line 78:
Please cite the precipitation and % snow. Is this from the same study (Spencer et al., 2019) which is cited in the discussion in reference to the sub surface storage capacity of the watersheds?

Line 83:
"Talus slopes" Please expand this description to include more information on the relative size of this geographic feature in the upper watersheds. Previous studies of source waters to alpine watersheds in the Rocky Mountains (suggested references listed below) indicate that talus slopes and underlying features can be significant source water areas.

Is there any information or indication of permafrost, ice lenses, or rock glaciers in the alpine talus areas that could provide a unique source water?

Caine, N, 2010. Recent hydrologic change in a Colorado alpine basin: an indicator of permafrost thaw?
https://doi.org/10.3189/172756411795932074

Clow, D. W., Schrott, L., Webb, R., Campbell, D. H., Torizzo, A., and Dorblaser, M.: Ground water occurence and contributions to streamflow in an alpine catchment, Colorado Front range, Ground Water, 41, 937–950, 2003.

Hood, J. L., Roy, J. W., and Hayashi, M.: Importance of groundwater in the water balance of an alpine headwater lake, Geophys. Res. Lett., 33, L13405, doi:10.1029/2006GL026611, 2006.

Roy and Hayashi, 2009. Multiple, distinct groundwater flow systems of a single moraine-talus feature in an alpine watershed.
https://doi.org/10.1016/j.jhydrol.2009.04.018

Williams et al., 2006. Geochemistry and source waters of rock glacier outflow, Colorado Front Range.
**https://doi.org/10.1002/ppp.535**

Line 86:
Can the amount of glacial till deposits be estimated or quantified for the sub-watersheds? There is no indication of spatial extent beyond description on line 80. It would help the reader to understand the potential storage capacity of the till especially since till water was excluded as a potential source water due to sampling well contamination (line 181). One suggestion is moving the citation on line 444 (AGS, 2004) to section 2 study site description and elaborating on the description of the "spatially heterogenous surficial deposits…" to help describe the watershed(s) in more detail.

Line 95:
Figure 1. It would be helpful to define tree line (separation of alpine from forested area within the sub-watershed. Important because the paper is framed as a study related to "forest disturbance" so the alpine portion of the study areas should be clearly separated from the forested areas.

Also please add the locations of the seeps that were sampled and used as potential end members in EMMA.

Line 125:
Snowmelt collection methods. Perhaps expand explanation of the snowmelt sample timing in order to reduce known uncertainty of changes in snowmelt chemistry related to timing of the melt. There is a known ionic pulse at the initiation of snowmelt (see Williams et al., 2009), which can be followed by dilute meltwater.

Are there any occurrences of dust on other impurities in the snowpack in this region which could impact the snowmelt chemistry or the timing and magnitude of snowmelt? Dry deposition was mentioned for rain water collection (line 121) but not for snowmelt.

Line 171:
Is the data from the Hobo sensors used in this paper? If not then this method does not support the paper and should be removed.

Line 274:
Bedrock groundwater, "excluded as a source at the upper sites". Please explain how the groundwater seep used in SEU (line 313, figure 8b) was classified as having consistently cool GW temperatures, but was not considered to be a "bedrock groundwater source"?

Line 276:
Suggest replacement of "a couple samples" with a more quantitative description.

Figures 5-8:
Suggest a more detailed explanation of the hysteresis present in the stream water samples. One option is to place the day of year (DOY) on each sample so readers can decipher movement within months which are plotted as one color. For example in figure 7A are the September samples temporally migrating in the mixing space or are sample points randomly distributed?

Figures 7 and 8
SEU and SEL both appear to have an unidentified source water in October as the October samples plot further away from the identified potential end-members. A more detialied interpretation of this observation is recommended for the discussion?

Line 325:
Section 5.3
It is understood that you were not able to sample in the winter, however you state that sampling stopped "before fall rains" ( line ) in previous section and in this section the "end" of seasonal sampling is stated as "start of the next year's snow accumulation period" (line 326). Just want to be clear on the terms used to describe the end of seasonal study periods.

If precipitation is lumped by rain and snow how do you know which form of precipitation is influencing stream flow in which season? For example line 342, the stream is "more similar to precipitation in June and July" Is this recent precipitation from rain or assumed to be the lagged input of snowmelt from the previous winter?
What would be helpful is a hyetograph over the study period so reader has some better sense of when the annual precipitation occurs. Also is there a way to present the timing and magnitude of snowmelt? Figure 10 suggests that there are multiple snowmelt pulses in winter and spring, can this be elaborated in the description of site climate and hydrologic inputs?

Line 399:
"increases the concentration in water" should be "increases tracer concentrations in the soil water.." if you are speaking about the inverse of water chemistry "dilution" from snowmelt.

Line 429:
Please clarify "increases in stream water chemistry" to specify that you are speaking about tracer concentrations or "concentration of stream water ions" (line 450). Consistent terminology will help the flow of the manuscript.

Line 457:
Please provide citation for this statement "Excess water associated with forest disturbance would infiltrate into the subsurface". These assumed hydrologic dynamics should be discussed in more detail because there is potential for a varying hydrologic response from forest disturbance.
 For example, in a forested snowmelt dominated watershed the timing and magnitude of snowpack accumulation and ablation in relation to canopy cover/density dynamics may be variable depending on forest dynamics.
Sublimation rates on canopy snow interception (see Classen and Downy, 1995), and impacts of forest shading on radiative forcing on snowpack ablation could influence infiltration rates.

I would also suggest mention of rainfall intensity relative to infiltration capacity in forested vs alpine or disturbed areas.

Recommended references to review

Molotch *et al*., 2009. Ecohydrological controls on snowmelt partitioning in mixed-conifer sub-alpine forests
**https://doi.org/10.1002/eco.48**

Harpold *et al*., 2014. Soil Moisture response to snowmelt timing in mixed-conifer subalpine forests.
**https://doi.org/10.1002/hyp.10400**

Musselman *et al.,* 2012 Influence of canopy structure and direct beam solar irradiance on snowmelt rates in a mixed conifer forest.
https://doi.org/10.1016/j.agrformet.2012.03.011

Line 475:
Figure 10 caption revision. Second sentence is an interpretation of the graph rather than a description and should be included in the text. Recommend clarifying text description of "more responsive" and "slower recession slopes" in reference to depth to groundwater below the surface.

Figure 10: In the soil/till GW, what causes the sharp response (increase in water table elevation) in November?  Is this related to early season snowfall that melts or other factor such as vegetative senescence? Does the chemistry change in that water source in late fall?

Can you explain the two separate groundwater level increases in the till well that occur in February and then again in March/April? Is this related to intermittent snowpack throughout the winter (as briefly mentioned in the snowmelt sampling methods line 125)?

Line 480:
Replace "old water" with a more accurate description representative of transit time or sub-surface residence time rather than speaking to the age of the water, or define old water to mean "reacted" waters that have had extended contact time with the sub-surface (see Liu *et al*. 2004)  The same suggestion  was made previously for defining the use of "old water" in the abstract.

Line 485:
Indicates that till groundwater could be slowly released to the stream (longer recession in Figure 10).  It is not clear if the intention was to suggest that this could be the unidentified source water end member in late fall in Star East, but was not was not captured or used in EMMA due to experimental design issues leading to well contamination?

Line 486:
Please expand the conclusion/suggestion that till groundwater (although not used as an end member for EMMA) has the potential to mute the effects of disturbance on peak flow. I assume you are referring to forest disturbance, but it is not clear of the locational relationship between till groundwater sources and forested areas within the watersheds. Is the till groundwater believed to be sourced from direct overhead recharge (in the same location as currently existing forests)? or is there another hypothesized mechanism of recharge such as mountain block recharge from higher alpine regions already void of forest cover?

---

## Author Comment (AC1) · 12 May 2020

**Author response to Anonymous Referee #1:**
**Our replies to referee comments (black italics) are provided below in blue.**

*Anonymous Referee #1 comments:*

*General comment*

*This works uses hydrochemical data to describe and infer runoff generation processes in the subcatchments of the Rocky Mountains. The topic is certainly interesting for the readership of HESS. The manuscript is generally well written. However, there are two main points that do not sound convincing to me: i) the focus on catchment resilience and disturbance, that do not appear to be logically linked to the investigations carried out and sounds out of context; ii) the presence of hydrochemical data only: despite the powerful nature of hydrochemistry as hydrological tracer, the combination of racer data and hydrometric data can help to unravel the complexity of hydrological processes at the catchment scales. Thus, the manuscript fails to describe in a robust, quantitative, and convincing way how water moves through this landscape in response to both rainfall and snowmelt. As a result, a clear contribution of this study to the body of knowledge is not evident. Please, find some specific and minor comments below.*
**Reply:** The authors thank the referee for their comments.

i) Hydrological resilience observed in this region (e.g., Harder et al., 2015; Goodbrand and Anderson, 2016) was the motivation to undertake this research. Others have suggested that complex subsurface flow pathways and large subsurface storage are potential factors that lead to hydrologic resilience (Harder et al., 2015) but, critically, little is known about runoff generation in the eastern slopes of Alberta's Rocky Mountains. To address this evidence gap, developing a conceptualization of groundwater-surface water interactions and runoff generation processes was the first step towards understanding why this region appears to be resilient to change. Despite this, we appreciate the referee's concerns about the lack of linkage between resilience and the research presented in our draft paper. To address the concerns of the referee, the Introduction will be reformulated to focus on understanding runoff generation in regions with permeable bedrock and deep soils/glacial till because this was ultimately our intention. All references to watershed resilience will be removed as suggested.

ii) We agree with the referee's comment regarding coupled geochemical and hydrological data to unravel hydrologic behaviour at the watershed scale. Indeed, this study is part of a larger research project that has been published in part. Our first published manuscript describes precipitation-runoff and storage dynamics in Star Creek using hydrometric data and suggests a conceptualization of runoff generation at the end of the manuscript (Spencer et al. 2019). Another manuscript (currently in prep) will use additional lines of evidence to further link these factors to water table responses, hydrologic connectivity, and structural controls on stream water contributions in Star Creek. While the hydrometric analysis/results presented in Spencer et al. (2019) were able to inform some aspects of runoff generation in the study region, the component of our research presented in our submission to HESS is meant to help clarify their conceptualization of runoff generation. Some suggestions from Referee #1 to add hydrometric and water table data will be incorporated into this manuscript, but there will be limits on what

can be included based on the data already published in Spencer et al. (2019). We also worry that the addition of too much extra data will make the manuscript too dense, resulting in an overall loss of clarity (or less easily digestible by readers).

*Specific comments:*

*The abstract is a bit vague. The motivation sounds weak, there are no specific objectives, the methods are partly unclear (water sources were sampled for what kind of analysis?), and the concept of hydrological resilience is not specified. I suggest revising it entirely.*
**Reply:** The abstract will be heavily revised to reflect the changes in the Introduction and Discussion and to clarify the conclusions that were made in relation to runoff generation.

*- The Introduction fails to clearly stress what it is not well known about the specific topic and what is the main research gap, and the reader, at the end of the Introduction is left wondering why another study on streamflow contribution is needed. An overall objective and testable hypothesis is not reported. The two specific objectives are introduced quite abruptly, without a clear and logical connection with the paragraph above. I suggest to heavily revise the Introduction to keep these points into consideration.*
**Reply:** The draft Introduction will be expanded and revised to better clarify research gaps and the concomitant rationale for this study.

The eastern slopes of Canada's Rocky Mountains have complex geology/surficial geology composed of permeable, fractured sedimentary bedrock overlain by deep glacial till (3 m on average). Others have hypothesized that these complex subsurface flow pathways may be responsible for the muted response in streamflow following disturbance (Harder et al., 2015; Goodbrand and Anderson, 2016) but a bespoke conceptualization of runoff generation is needed. The local geology may control runoff generation and subsurface flow pathways in ways not consistent with broadly accepted paradigms of runoff generation applicable to regions with largely unfractured bedrock overlain by thinner surface materials. While there are many studies in regions with permeable bedrock or deep soils/glacial till, few exist in regions with both features in series. As a result, while some implications can be drawn from regions with either permeable bedrock or deep soils or till, more research is needed to conceptualize runoff processes in these systems.

*- 190-208. I suggest to consider the work by Barthold (2001) and to specify the reported approaches were preferred over this method. Moreover, briefly mention how TVR and LDA work to allow the reader better understanding the methods that were used.*
*https://agupubs.onlinelibrary.wiley.com/doi/pdf/10.1029/2011WR010604*
**Reply:** The draft text will be revised to specify why the approach used here was preferred over the method outlined in Barthold et al. (2011). The explanation of how TVR and LDA work will be expanded.

We are aware of the work by Barthold et al. (2011) and did consider the methods presented in the paper at the onset of this research. However, the stream water falls outside the bounds of the sources, which violates the EMMA assumption that there are no missing sources (Figures 7 and

8). This and the larger variability in source water than stream water (quantified by coefficient of variation) indicate that EMMA could not be run in its entirety to determine percent contributions from each stream water source. The inability to run the unmixing routine hindered the used of the methods outlined in Barthold et al. (2011) because the second criteria in the automated procedure requires running many iterations of source water contributions.

Other methods that could be used to indicate whether sources were well separated and if tracers showed minimal variation were evaluated. TVR and LDA have been presented as effective parameters to subjectively determine if tracers are included in the analysis and if sources are well separated or grouped appropriately (Pulley et al., 2015; Pulley and Collins, 2018; and others – see comprehensive review in Collins et al., 2017 – Journal of Environmental Management). These methods have been automated in the SIFT (SedIment Fingerprinting Tool) open source R shiny software described in Pulley and Collins (2018). We used this portion of the SIFT routine.

*- I suggest merging Figs. 5 and 6 (making a multi-panel figure) and sections 5.2.1 ad 5.2.2, and Fig. 7 and 8 and sections 5.2.3 and 5.2.4 in order to present the results from the two subcatchments more organically. Similarly, I recommend merging Section 5.3.1 and 5.3.2 (Star West), and 5.3.3 and 5.3.4 (Star East) to avoid too much text and results in fragmentation.*
**Reply:** We agree that this will streamline the Results and Discussion and will revise as suggested.

*- Since there is, at least in some cases, a strong seasonal pattern in hydrochemistry, I suggest considering making a time series plot of the different water sources in the two subcatchments in order to show, for instance, when and to which extent the stream water signature gets closer to that of hillslope groundwater and riparian water. In addition, the Authors might consider adding times series of groundwater temperatures or boxplots, as this tracer is part of the story and was shown to be able to partly explain groundwater contributions to streamflow.*
**Reply:** We will consider adding a time series plot of water sources and stream water as suggested. These plots would show the patterns we are describing in the insets in Figures 7 and 8 with more clarity. They will also help with other discussion points made here (as indicated in reply to the comment below: line 417-418) and by Referee #2.

*- 417-418. Which evidence do the Authors have to infer the temporal dynamics of hillslope water moving to the stream? Moreover, how could the Authors describe old water mobilization without having quantified its proportion in stream water? Or this is a general statement not based on the presented dataset? Please, explain.*
**Reply:** Temporal dynamics of hillslope water moving to the stream are inferred from the PCA plots as stream water is similar to the various source water at different times of the year. There is also evidence in our time series of stream water, where concentrations of some chemical constituents increase in the early spring just as snowmelt starts, which appears to be similar to the piston flow observed in other watersheds. Text and time series plots will be added to clarify this point.

*- 464-474. I feel this part is quite out-of-context and disconnected from the previous discussion. In general, I think that focusing on catchment resilience is not so straightforward and sound a*

*bit contrived to me. The same comment applies to the Conclusions.*
**Reply:** All references to catchment resilience will be removed as suggested.

*Minor comments and technical corrections:*

*1. The title is long and complex. I suggest making it more compact and clearer.*
**Reply:** Revision of the title will be considered.

*11. I suggest to change as follows: "A lack of : : :but mechanisms governing: : :".*
**Reply:** Sentence will be revised as suggested.

*13. ": : :although much: : :": I cannot see the logical link in this sentence. Please revise.*
**Reply:** The abstract will be revised for clarification and to reflect the changes in the Introduction and additions to the draft manuscript.

*13-14. "to interpret how forest disturbance may impact streamflow quantity". I would not focus on understanding runoff generation processes to this aim, but mostly on the ecohydrological role of forest on streamflow. Please, revise.*
**Reply:** The focus of the manuscript will be revised. See reply to general comment and Introduction comment above. The abstract will be revised for clarification and to reflect the changes in the Introduction and additions to the draft manuscript.

*22. "but was unlike the measured sources": this sentence is not clear before reading the abstract. Please, clarify.*
**Reply:** This statement will be clarified.

*29: Perhaps put it more general, mentioning pathogens.*
**Reply:** This will be revised as suggested.

*35. What do the Authors refer to by "features"? Please explain.*
**Reply:** This will be revised for clarification: "features" was referring to "watershed features (e.g., bedrock, surficial geology, wetlands)" and will be added for clarification.

*112. ": : :a priori: : :": Was there any evidence, field observation, previous study or knowledge of the area that allowed for this assumption?*
**Reply:** Stream water sources were hypothesized based on field observations and previous knowledge of the area. This research is part of the Southern Rockies Watershed Project, which has been conducting research in this watershed and other watersheds in the area since 2004. As such, the stream water sources were based on local knowledge from working in these mountains. This will be clarified in the text.

*193. TVR: please report the definition and possibly the equation to let the reader immediately understand it.*
**Reply:** The text will be re-arranged and the definition and equation will be stated more explicitly so the reader can immediately understand it.

*229-230. This sentence is not clear to me (without reading the cited references). Please specify.*
**Reply:** Sentence will be revised for clarification.

*245. leu?*
**Reply:** This is a typo and will be corrected.

*269: Perhaps add "compared to bedrock groundwater".*
**Reply:** We prefer not to change this sentence as it is a stand-alone statement. There was no clear temporal pattern observed… not compared to the other sources. However, we will revise the text to ensure our meaning is clear.

*Fig. 6b). Could the Authors perhaps colour-code samples for season (spring, summer, fall)?*
**Reply:** Colour-coding for season will be considered. However, with the addition of time series as suggested above, the new time series will likely be referenced and insets removed. Time series will show the same pattern the authors are pointing out here, making the insets redundant.

*322. Which are these months?*
**Reply:** Months will be added to the text: '…months of open-water flow (Apr-Oct)…'

*340. Why a source might be missing? Please, explain.*
**Reply:** Text will be added for clarification: '…because some samples fell outside of the mixing space defined by the mean and standard deviation of the sampled sources…' This logic is well established in the international literature.

*393-394: Are groundwater levels available? Their temporal patterns could help understand which feeds which. Perhaps some piezometers could be installed for a follow-up study.*
**Reply:** Groundwater levels are available in the riparian, toe slope and upper hillslope areas. We will explore this suggestion and include this comparison in revisions if this helps strengthen the revised narrative.

*429-430. What does "increase in stream water chemistry" mean? Moreover, how would be possible to infer connectivity through hydrochemical data only? Some speculations could be done but a combination of hydrometric and tracer data would serve this purpose better.*
**Reply:** This will be revised to "increase in stream water ion concentrations".

These water chemistry observations are taken in conjunction with observations published in Spencer et al., 2019 and another manuscript that is currently in prep. The other manuscripts contain hydrometric, meteorological, and groundwater data that help infer connectivity.

Text will be revised to clarify that these inferences are being made in conjunction with other studies that were carried out in the same watersheds during the same time as in the current study.

*431. Contributions to what? Please specify.*
**Reply:** Text will be revised to: 'Source water contributions to the stream…'

*433. It cannot be all rain water, can it? Please, revise/explain.*
**Reply:** No, this is not all rain water. It is almost entirely snowmelt as this is a snow-dominated

watershed but some summer rain storms would also contribute to runoff. Snowmelt saturates the landscape in May and causes a significant dilution effect in the stream. However, the water that is contributing to the stream is not as dilute as snowmelt itself, suggesting that there is a mixture of snowmelt and hillslope water contributing to the stream. We state that it is most like precipitation to stress this dilution effect, not to suggest that the water entering the stream is pure snowmelt or rain. Water chemistry of rainfall and snowmelt are essentially identical so there is no way to separate the contribution of rain and snow.

Our draft text will be modified to clarify these points.

*Possible useful readings for additional analyses and for the discussions section:*
*Correa, A., Breuer, L., Crespo, P., Célleri, R., Feyen, J., Birkel, C., Silva, C., Windhorst, D., 2019. Spatially distributed hydro-chemical data with tempo-rally high-resolution is needed to adequately assess the hydrological functioning of headwater catchments. Science of The Total Environment 651, 1613–1626. https://doi.org/10.1016/j.scitotenv.2018.09.189*
*Godsey, S.E., Hartmann, J., Kirchner, J.W., 2019. Catchment chemostasis revisited: Water quality responds differently to variations in weather and climate. Hydrological Processes 33, 3056–3069. https://doi.org/10.1002/hyp.13554*
*Hoeg, S., Uhlenbrook, S. and Leibundgut, C., 2000. Hydrograph separation in a mountainous catchment - combining hydrochemical and isotopic tracers. Hydrol. Process., 14: 1199-1216. doi:10.1002/(SICI)1099-1085(200005)14:7<1199::AIDHYP35>3.0.CO;2-K*
*Hrachowitz, M., Bohte, R., Mul, M.L., Bogaard, T.A., Savenije, H.H.G., Uhlenbrook, S., 2011. On the value of combined event runoff and tracer analysis to improve understanding of catchment functioning in a data-scarce semi-arid area. Hydrol. Earth Syst. Sci. 15, 2007–2024. https://doi.org/10.5194/hess-15-2007-2011*
*Nadal-Romero, E., Khorchani, M., Lasanta, T., García-Ruiz, J.M., 2019. Runoff and Solute Outputs under Different Land Uses: Long-Term Results from a Mediterranean Mountain Experimental Station. Water 11, 976. https://doi.org/10.3390/w11050976*
*Penna, D., van Meerveld, H.J., Zuecco, G., Dalla Fontana, G., Borga, M., 2016. Hydrological response of an Alpine catchment to rainfall and snowmelt events. Journal of Hydrology 537, 382–397. https://doi.org/10.1016/j.jhydrol.2016.03.040*
*Suecker, J.K., Ryan, J.N., Kendall, C., Jarrett, R.D., 2000. Determination of hydrologic pathways during snowmelt for alpine/subalpine basins, Rocky Mountain National Park, Colorado. Water Resour. Res. 36, 63–75. https://doi.org/10.1029/1999WR900296*

**Reply:** Thank you for these suggestions. These references will be incorporated where applicable.

---

## Author Comment (AC2) · 12 May 2020

**Author response to Anonymous Referee #2:**
**Our replies to referee comments (black italics) are provided below in blue.**

*Anonymous Referee #2 comments:*

*Overview*

*This is an interesting research manuscript with well-defined objectives of using EMMA to understand relative variation in stream water sources in forested and alpine watersheds of the Canadian Rocky Mountains. Overall the manuscript is sound and has relatively minor grammatical errors. However, it is suggested that the introduction and discussion sections more clearly identify the secondary objectives and how the study . It is unclear if the results of this study have a primary goal of improving understanding of the variability between different groundwater sources of (e.g. bedrock groundwater vs. glacial till groundwater) in alpine and sub-alpine watersheds, or to understand/predict how future forest disturbance will impact watershed hydrology and runoff generation.*
**Reply:** Firstly, we thank the referee for their positive comments.

Referee #1 has suggested that any discussion of hydrologic resilience be removed. We agree that revising the draft Introduction will strengthen the context of this research and clarify that the impetus to conduct this study was to improve our understanding of runoff generation in watersheds with permeable fractured bedrock and deep glacial till (compared to impermeable bedrock and shallow soils, like in some research areas). The primary goal was to improve understanding of variability between different groundwater sources, rather than predicting how future forest disturbance will impact watershed hydrology and runoff generation.

*It is suggested that the study site description be expanded to more clearly quantify the size and extent of surface and sub-surface biological and geological features (talus slopes, alpine areas, glacial till, forested areas, riparian areas, etc.). It was hard to determine where and why the division of upper and lower sub-watersheds was chosen for this study and how that division was critical to addressing the question of the impacts of forest disturbance on hydrologic disturbance. Both upper and lower sub-watersheds had forested areas so the separation of upper and lower sub-watersheds did not appear to be a proxy for forested vs unforested/disturbed areas.*
**Reply:** The separation between upper and lower watersheds was made in 2008 in anticipation of the Mountain Pine Beetle expanding into our research watersheds. The upper stations were positioned at the transition between 1) primarily lodgepole pine dominated forests and 2) the narrower band of subalpine fir dominated stands and the treeless alpine valley above this. The higher elevation sub-watersheds would have theoretically been spared from the pine beetle attack allowing for a comparison after beetle attack. Pine beetle did not reach our research watersheds, so they were maintained as reference watersheds. The upper sites were maintained because they represent this difference between subalpine/alpine zone and the upper montane zone. The upper montane zone is fully forested, and slopes are slightly gentler. The subalpine/alpine zone is partly forested but also has regions of talus slopes, near vertical bedrock cliffs, and alpine grasses. The streams in the alpine flow over bedrock in some places with little colluvial or alluvial material. Although both are "forested", the upper site is only partly forested. The comparison here comes from the conceptual diagram of runoff generation in Star Creek from Spencer et al. (2019) where the authors suggested that these two regions process water differently, have different amounts of subsurface storage, and affect the stream hydrograph at the watershed outlet in different ways. Thus, rather than being a comparison of forested and

unforested sub-watersheds, the research presented here is meant to advance the conceptualization from Spencer et al. (2019), comparing subalpine/alpine to upper montane responses.

The draft study description will be expanded to clarify the extent of subsurface features and differences between the upper and lower sub-watersheds. The objectives will also be clarified as indicated above.

*Secondly, the discussion and conclusion sections should be expanded to more clearly indicate how the results in this study advance or improve the scientific understanding of "how forest disturbance may impact streamflow quantity". It is unclear if the study was able to confirm or reject the hypothesis stated in the abstract that "slow release of groundwater from glacial till" (line 24) generates "hydrologic resilience" in the Rocky Mountains?*
**Reply:** Referee #1 suggested that the Discussion and Conclusion sections relating to hydrologic resilience in the Rocky Mountains be removed. As such, this section will no longer discuss how forest disturbance may impact streamflow quantity. The hypothesis that there is a slow release of groundwater from glacial till will be revised to reflect this change. Clarification on the role of groundwater from glacial till will also be added to the revised manuscript.

*Line 18: Suggest defining "old water" as related to time the water has spent in the watershed rather than the true age of water.*
**Reply:** Text will be revised as suggested.

*Line 20: In Star east in September and October the stream water was unlike the sources. What is the additional source or is it a mixed signal?*
*Same statement is in the conclusion but the proposed explanation of the "missing" sources is either absent or not clear in the conclusion. Clarification would be helpful.*
**Reply:** In part, the confusion comes from the fact that it is not entirely clear to us what the additional source is. We can infer from water temperature in seeps that it is likely a bedrock source, but some questions remain. Text will be added to the draft Discussion and Conclusion to clarify our postulations about the missing source.

*Line 29: What is the specific reference for beetle infestation? A few Studies from the Rocky Mountains to consider reviewing, Pugh & Small, 2011. https://doi.org/10.1002/eco.239*
*Bearup et al., 2014. https://doi.org/10.1038/nclimate2198*
**Reply:** Boon (2012) is the specific reference for beetle infestation. The authors will incorporate Pugh & Small (2011) and Bearup et al. (2014) as suggested.

*Line 53: Consider reference of Cowie et al. (2017) here as that study does use EMMA to examine potential source waters from bedrock groundwater, glacial till groundwater, talus slope water, and soil water on streamflow contributions in forested and alpine watersheds in the Rocky Mountains.*
**Reply:** Cowie et al. (2017) was not initially included in this section of the draft Introduction because bedrock in the Colorado Rocky Mountains is mainly granodiorite, rather than permeable sedimentary bedrock and we were trying to make a distinction between bedrock types. However, Cowie et al. (2017) is relevant to the current knowledge and overall discussion in this paper. Although bedrock is not as permeable as the sedimentary bedrock present in Alberta's Rocky Mountains, it is still shown to be a source of streamflow, thus, it will be incorporated into the revised Introduction as suggested.

*Line 77: Define area weighted precipitation. Was precipitation measured at multiple elevations? With > 1000m elevation change how much does the total precipitation change over that gradient? One*

*suggestion is use of a hypsometric curve to distribute precipitation over elevation (see Cowie et al., 2017)*
**Reply:** Precipitation was measured at nine precipitation gauges in Star Creek and a neighbouring watershed (North York Creek). Spencer et al. (2019) used the thiessen area-weighted method to estimate average watershed precipitation rather than using one particular rain gauge to represent the entire watershed. The area-weighted method is a good approximation of average precipitation if rain gauges are distributed at a range of elevations. Our gauges are distributed between 1482 m and 1873 m. Vertical headwalls and talus slopes in the alpine basins limit the presence of precipitation gauges above 1900 m. While hypsometric curves can be useful to display the change in precipitation over a gradient, it would not add to the current story in this manuscript.

Clarification will be added to the draft manuscript to define area weighted precipitation and a citation to Spencer et al. (2019) will also be added.

*Line 78: Please cite the precipitation and % snow. Is this from the same study (Spencer et al., 2019) which is cited in the discussion in reference to the sub surface storage capacity of the watersheds?*
**Reply:** Yes, this is the same data from Spencer et al. (2019), although 2015-2018 were added to the years of record. Spencer et al. (2019) will be cited as suggested.

*Line 83: "Talus slopes" Please expand this description to include more information on the relative size of this geographic feature in the upper watersheds. Previous studies of source waters to alpine watersheds in the Rocky Mountains (suggested references listed below) indicate that talus slopes and underlying features can be significant source water areas.*
*Is there any information or indication of permafrost, ice lenses, or rock glaciers in the alpine talus areas that could provide a unique source water?*
*Caine, N, 2010. Recent hydrologic change in a Colorado alpine basin: an indicator of permafrost thaw? https://doi.org/10.3189/172756411795932074*
*Clow, D. W., Schrott, L., Webb, R., Campbell, D. H., Torizzo, A., and Dorblaser, M.: Ground water occurence and contributions to streamflow in an alpine catchment, Colorado Front range, Ground Water, 41, 937–950, 2003.*
*Hood, J. L., Roy, J. W., and Hayashi, M.: Importance of groundwater in the water balance of an alpine headwater lake, Geophys. Res. Lett., 33, L13405, doi:10.1029/2006GL026611, 2006.*
*Roy and Hayashi, 2009. Multiple, distinct groundwater flow systems of a single moraine-talus feature in an alpine watershed. https://doi.org/10.1016/j.jhydrol.2009.04.018*
*Williams et al., 2006. Geochemistry and source waters of rock glacier outflow, Colorado Front Range. https://doi.org/10.1002/ppp.535*
**Reply:** Talus slopes terminate in the alpine and forested regions of the watershed. Streams or tributary features flowing from the talus slopes have not be observed. Snowmelt and rain may be temporarily stored in talus slopes as documented in other Rocky Mountain watersheds (Cowie et al., 2017; Clow et al. 2003; Hood and Hayashi et al., 2015; McClymont et al., 2010), but it is likely that this water would infiltrate into the subsurface prior to arriving in the stream, thereby changing the chemical concentrations in this water. Permafrost, ice lenses, or rock glaciers are not present in the alpine talus areas based on the data we have from the Alberta Geologic Society, so it is unlikely that they could serve as a potential unidentified source.

The description of geographic features will be expanded so differences between Star Creek and other Rocky Mountain watersheds (indicated above) can be better understood by the reader.

*Line 86: Can the amount of glacial till deposits be estimated or quantified for the sub-watersheds? There is no indication of spatial extent beyond description on line 80. It would help the reader to understand the potential storage capacity of the till especially since till water was excluded as a potential source water*

*due to sampling well contamination (line 181). One suggestion is moving the citation on line 444 (AGS, 2004) to section 2 study site description and elaborating on the description of the "spatially heterogenous surficial deposits…" to help describe the watershed(s) in more detail.*

**Reply:** The description of geologic features will be expanded in the revised site description. Data from the Alberta Geologic Society can be used to estimate the extent of glacial till, talus slopes, and other surficial deposits.

*Line 95: Figure 1. It would be helpful to define tree line (separation of alpine from forested area within the sub-watershed. Important because the paper is framed as a study related to "forest disturbance" so the alpine portion of the study areas should be clearly separated from the forested areas.*
*Also please add the locations of the seeps that were sampled and used as potential end members in EMMA.*

**Reply:** Although the way the paper is being framed is changing in response to comments from Referee #1, adding the extent of the forested area will help readers visualize the watershed. The extent of the forested area will be added to Figure 1.
Seep locations will be added to Figure 1.

*Line 125: Snowmelt collection methods. Perhaps expand explanation of the snowmelt sample timing in order to reduce known uncertainty of changes in snowmelt chemistry related to timing of the melt. There is a known ionic pulse at the initiation of snowmelt (see Williams et al., 2009), which can be followed by dilute meltwater.*

**Reply:** Snowmelt collection methods will be expanded but in general, snowmelt samples were collected at random and opportunistically when our field crew happened to be on site rather than at specific intervals. A time series of source water will also be added to the manuscript (suggested by Referee #1), which should identify the presence of a pulse in ion concentration at the onset of snowmelt if it was captured by the sampling campaign. There were some samples collected in early May that had elevated concentrations of ions compared to samples collected in early June, but ion concentrations were still far lower than concentrations observed in soils. The known ionic pulse in snowmelt to which Referee #2 is referring is very interesting and we will investigate this further.

*Are there any occurrences of dust on other impurities in the snowpack in this region which could impact the snowmelt chemistry or the timing and magnitude of snowmelt? Dry deposition was mentioned for rain water collection (line 121) but not for snowmelt.*

**Reply:** Dry deposition of dust/dirt can be a problem in the summer when the landscape is directly exposed to wind but is not an issue in the winter when the ground is frozen and snow covered. Dry deposition from major cities and industrial areas is not known to be a problem because neither are in proximity to the study site. However, organic material shed from forest vegetation and excreted from wild animals would be deposited onto the snowpack.

*Line 171: Is the data from the Hobo sensors used in this paper? If not then this method does not support the paper and should be removed.*

**Reply:** Data from Hobo sensors were used in Figure 10 and to determine the temperature range of bedrock and till groundwater in wells.

*Line 274: Bedrock groundwater, "excluded as a source at the upper sites". Please explain how the groundwater seep used in SEU (line 313, figure 8b) was classified as having consistently cool GW temperatures, but was not considered to be a "bedrock groundwater source"?*

**Reply:** We were hesitant to classify the groundwater seeps as "bedrock groundwater" with moderate certainty because although the water was consistently cool, the water chemistry and

temperature range was not the same as the water sampled from the bedrock well. Groundwater seeps were 2 °C cooler than temperatures in the bedrock well and the ion concentrations in the seep groundwater were more variable than those in the groundwater well. A better explanation of groundwater seeps will be added to the draft text.

*Line 276: Suggest replacement of "a couple samples" with a more quantitative description.*
**Reply:** Will revise as suggested.

*Figures 5-8: Suggest a more detailed explanation of the hysteresis present in the stream water samples. One option is to place the day of year (DOY) on each sample so readers can decipher movement within months which are plotted as one color. For example in figure 7A are the September samples temporally migrating in the mixing space or are sample points randomly distributed?*
**Reply:** We will explore this suggestion and include this in revisions if it clarifies this issue without creating too much clutter. Adding a Julian day is an option but there are two years of data in this figure so it might add more confusion. Time series (as suggested by Referee #1) of stream and source water ion concentration would also help decipher movement at a finer scale. At the very least, more details will be added to the explanation in the Results to improve understanding of the patterns presented.

*Figures 7 and 8: SEU and SEL both appear to have an unidentified source water in October as the October samples plot further away from the identified potential end-members. A more detailed interpretation of this observation is recommended for the discussion?*
**Reply:** More details will be added to the Discussion as suggested.

*Line 325: Section 5.3: It is understood that you were not able to sample in the winter, however you state that sampling stopped "before fall rains" ( line ) in previous section and in this section the "end" of seasonal sampling is stated as "start of the next year's snow accumulation period" (line 326). Just want to be clear on the terms used to describe the end of seasonal study periods.*
**Reply:** The only sampling that was conducted 'before fall rains' was for groundwater seeps. Groundwater seeps were sampled during three sampling campaigns to target peak flows, recession flow, and baseflow (a range in watershed "wetness"). The final sampling campaign at the end of the summer (before fall rains) was an attempt to capture the clearest signal of "true" groundwater from the seeps if they were influenced at all by snowmelt or rainfall through the spring and summer. All other samples for other sources and the stream were collected from April to October (the start of the next water year). We will ensure that the language used in the revised manuscript is consistent and clear so there is no confusion for the reader.

*If precipitation is lumped by rain and snow how do you know which form of precipitation is influencing stream flow in which season? For example line 342, the stream is "more similar to precipitation in June and July" Is this recent precipitation from rain or assumed to be the lagged input of snowmelt from the previous winter?*
*What would be helpful is a hyetograph over the study period so reader has some better sense of when the annual precipitation occurs. Also is there a way to present the timing and magnitude of snowmelt? Figure 10 suggests that there are multiple snowmelt pulses in winter and spring, can this be elaborated in the description of site climate and hydrologic inputs?*
**Reply:** Ion concentrations of rainfall and snowmelt are essentially identical so there is no way to decipher between these two sources based on the chemistry we have (we do not have isotopes). Thus, the lagged input of snowmelt and recent rainfall cannot be separated. Further, the 2-week sampling schedule does not allow for the resolution needed to really identify a rainfall pulse moving through the watershed. A hyetograph could be combined with a hydrograph and

continuous snow depth data to show these dynamics in part and help the reader visualize the climatic conditions. Spencer et al. (2019) showed that snowmelt was the main period of hydrologic connectivity and that there was a larger streamflow response to rainfall events closer to the snowmelt period than later in the summer. Many rainfall events in the summer did not show a response in shallow groundwater wells suggesting that most percolated to deeper layers. These points and others suggested above will be added to description of climate and hydrology and to the revised manuscript where discussed to clarify these issues for the reader.

*Line 399: "increases the concentration in water" should be "increases tracer concentrations in the soil water.." if you are speaking about the inverse of water chemistry "dilution" from snowmelt.*
**Reply:** Will be revised as suggested.

*Line 429: Please clarify "increases in stream water chemistry" to specify that you are speaking about tracer concentrations or "concentration of stream water ions" (line 450). Consistent terminology will help the flow of the manuscript.*
**Reply:** Will be revised to 'an increase in stream water ion concentrations…'.

*Line 457: Please provide citation for this statement "Excess water associated with forest disturbance would infiltrate into the subsurface". These assumed hydrologic dynamics should be discussed in more detail because there is potential for a varying hydrologic response from forest disturbance.*
*For example, in a forested snowmelt dominated watershed the timing and magnitude of snowpack accumulation and ablation in relation to canopy cover/density dynamics may be variable depending on forest dynamics. Sublimation rates on canopy snow interception (see Classen and Downy, 1995), and impacts of forest shading on radiative forcing on snowpack ablation could influence infiltration rates. I would also suggest mention of rainfall intensity relative to infiltration capacity in forested vs alpine or disturbed areas. Recommended references to review:*
*Molotch et al., 2009. Ecohydrological controls on snowmelt partitioning in mixed-conifer sub-alpine forests. https://doi.org/10.1002/eco.48*
*Harpold et al., 2014. Soil Moisture response to snowmelt timing in mixed-conifer subalpine forests. https://doi.org/10.1002/hyp.10400*
*Musselman et al., 2012 Influence of canopy structure and direct beam solar irradiance on snowmelt rates in a mixed conifer forest. https://doi.org/10.1016/j.agrformet.2012.03.011*
**Reply:** The discussion on hydrologic resilience will be removed as suggested by Referee #1.

*Line 475: Figure 10 caption revision. Second sentence is an interpretation of the graph rather than a description and should be included in the text. Recommend clarifying text description of "more responsive" and "slower recession slopes" in reference to depth to groundwater below the surface.*
**Reply:** Figure caption will be revised and clarification of these terms will be added.

*Figure 10: In the soil/till GW, what causes the sharp response (increase in water table elevation) in November? Is this related to early season snowfall that melts or other factor such as vegetative senescence? Does the chemistry change in that water source in late fall?*
*Can you explain the two separate groundwater level increases in the till well that occur in February and then again in March/April? Is this related to intermittent snowpack throughout the winter (as briefly mentioned in the snowmelt sampling methods line 125)?*
**Reply:** Figure 10 was added simply to characterize the water table recession and infer the conductivity of the bedrock well compared to the glacial till well. The specifics of the responses in November, February, and March/April were not investigated in part because these responses were for 2017 and we were focusing on 2014/2015 seasons. However, we will look into the

glacial till water table/snowmelt dynamics to address this comment and determine if more information on runoff generation can be inferred from these data.

*Line 480: Replace "old water" with a more accurate description representative of transit time or sub-surface residence time rather than speaking to the age of the water, or define old water to mean "reacted" waters that have had extended contact time with the sub-surface (see Liu et al. 2004) The same suggestion was made previously for defining the use of "old water" in the abstract.*
**Reply:** Any reference to "old water" will be defined and revised to mean "reacted" water or water that was stored in the watershed over winter rather than a specific age of the water.

*Line 485: Indicates that till groundwater could be slowly released to the stream (longer recession in Figure 10). It is not clear if the intention was to suggest that this could be the unidentified source water end member in late fall in Star East, but was not was not captured or used in EMMA due to experimental design issues leading to well contamination?*
**Reply:** There is a possibility that till groundwater is the unidentified source in late fall in Star East; however, there is no concrete evidence presented here that allows us to make this conclusion. This would simply be speculation based on other lines of evidence and observations made in other regions of North America. This section will be clarified and any direct inferences we can make will be added.

*Line 486: Please expand the conclusion/suggestion that till groundwater (although not used as an end member for EMMA) has the potential to mute the effects of disturbance on peak flow. I assume you are referring to forest disturbance, but it is not clear of the locational relationship between till groundwater sources and forested areas within the watersheds. Is the till groundwater believed to be sourced from direct overhead recharge (in the same location as currently existing forests)? or is there another hypothesized mechanism of recharge such as mountain block recharge from higher alpine regions already void of forest cover?*
**Reply:** Reference to resilience will be removed from the conclusions (and the rest of the manuscript) as suggested by Referee #1. However, clarification of till groundwater responses/sources will be added to the manuscript.

---

## Editor Comment (EC1) · Bettina Schaefli (Editor) · 14 May 2020

This reviewer is very critical about the scientific (added) value of the paper and recommends rejection in the formal reviewer report.

This overall assessment is motivated by the statement that

"the manuscript fails todescribe in a robust, quantitative, and convincing way how water moves through this landscape in response to both rainfall and snowmelt.". This statement is motivated by two main critics:

"i) the focus on catchment resilience and disturbance, that do not appear to be logically linked to the paper investigations carried out and sounds out of context;

ii) the presence of hydrochemical data only: despite the powerful nature of hydrochemistry as hydrological tracer,the combination of racer data and hydrometric data can help to unravel the complex-ity of hydrological processes at the catchment scales."

The authors' response to the second point does not contain any detailed plans at this stage. I am waiting for a third editor report but I already invite the authors to provide a more detailed planned how you plan to address the above second point before I can give a recommendation on the submission of a revised version.

---

## Author Comment (AC3) · 19 May 2020

**Author response to Editor comment:**
**Our replies to editor comments (black italics) are provided below in blue.**

*This reviewer is very critical about the scientific (added) value of the paper and recommends rejection in the formal reviewer report. This overall assessment is motivated by the statement that "the manuscript fails to describe in a robust, quantitative, and convincing way how water moves through this landscape in response to both rainfall and snowmelt.".*

*This statement is motivated by two main critics:*

*"i) the focus on catchment resilience and disturbance, that do not appear to be logically linked to the paper investigations carried out and sounds out of context;*

*ii) the presence of hydrochemical data only: despite the powerful nature of hydrochemistry as hydrological tracer, the combination of racer data and hydrometric data can help to unravel the complexity of hydrological processes at the catchment scales."*

*The authors' response to the second point does not contain any detailed plans at this stage. I am waiting for a third editor report but I already invite the authors to provide a more detailed planned how you plan to address the above second point before I can give a recommendation on the submission of a revised version.*

**Reply:** We thank the editor for their comment and clarification on this issue.

We agree that hydrometric data are an important part of this story. The published manuscript we refer to in our first reply (Spencer et al., 2019) includes estimates of dynamic storage by water balance method, a hydrograph recession analysis to infer groundwater contributions, snowfall-runoff ratio relationships to understand the influence of pre-winter storage, and hydrograph response (event rise) at the storm scale in wet and dry seasons. These analyses were used to develop a conceptual model of storage and runoff generation for alpine and subalpine/montane regions in Star Creek. This was the first step in understanding runoff generation in regions with glacial till overlaying permeable sedimentary bedrock. The aim of this draft manuscript submitted to HESS is to develop further lines of evidence to address the complexity of runoff generation in this region. While Harder et al. (2015) postulated that this region has complex subsurface flow pathways, to our knowledge, no studies have characterized runoff generation in the subalpine region of Alberta's Rocky Mountains. The Introduction of the present draft manuscript will be reformulated to stress the connection with Spencer et al. (2019) and position this research in context with other regions with shallow soils and impermeable bedrock.

A figure (such as the draft in Figure 1 below) will be added to address the lack of hydrometric data in the draft manuscript. It was an oversight not to include any data that shows the runoff patterns in the draft manuscript. This figure will help show the link between observed inputs (daily precipitation and continuous snow depth) to observed responses (specific discharge, stream water chemistry, and shallow groundwater levels). It will also address other comments/concerns from both referees. Similar figures are often presented along with hydrochemical data in other publications (e.g., Blumstock et al., 2015; Barthold et al., 2017; Ali et al., 2010; Cowie et al., 2017; Inamdar et al., 2013; Hoeg et al., 2000; Sueker et al., 2000; Correa et al., 2019). A table with precipitation event characteristics is also often presented in conjunction with the hydrographs; however, storms are not the focus of our study so other metrics such as average annual discharge, percent of streamflow that occurs from May to July, and annual peak flow will also be added to further characterize the hydrological setting.

[Figure]

Figure 1. Draft of a figure that will be added to the draft manuscript. It is a work in progress but we wanted to attach it as an example of what we would like to include.

We would prefer not to include the unmixing model results because of the violation of multiple EMMA assumptions. We had explored this option before and decided that the errors associated with the percent contributions outweighed the contributions. This is similar to Hoeg et al., (2000) who found geographic hydrograph separations did not lead to conclusive results, and Inamdar et al., (2013), who suggested that caution should be applied when calculating percent contributions where there is large variation in end members and assumptions are violated. If the critical assumptions of EMMA were not violated, we would also have examined model accuracy using virtual mixtures of the sampled water sources since model tests using such mixtures are becoming more commonplace in the international literature using source apportionment methods. However, as we state above, assumption violation precludes inclusion of such work.

---

## Referee Comment (RC3) · Anonymous Referee #3 · 8 Jun 2020

**Summary of the paper:**

In this study, multiple tracers were used to identify dominant runoff generation mechanisms over two hydrologic years in Star-east and Star-west watersheds. Principal component analysis was used to reduce the complexity that may arise by analyzing every tracer combination. The study concluded that streamflow during early melt was dominated by hillslope groundwater. As snowmelt peaked, the entire landscape became connected and all the water sources contributed to streamflow (the proportion from different sources is not computed). During the Fall season, hillslope and bedrock groundwater became the major sources of streamflow in Star West watershed (proportions not computed), however the sources were unresolved in the Star East watershed. The authors then went on to conclude the subsurface flow pathways in this region are complex and this complexity along with slow release of groundwater from glacial till ensures hydrologic resilience in this region.

This study tries to resolve the seasonal sources of streamflow which is a very interesting research topic and definitely fit for this journal. However, quantitative estimates of source proportions are missing from this study which is possible to compute given the number of tracer variables that were monitored.

**Major comments:**
1. The abstract and introduction talks in detail about the concept of hydrologic resilience, however I do not find any attempt to quantify this statistic in the remainder of this article (except a very brief discussion on recession rates at the end). I will recommend either quantifying resilience or removing it (at least from the abstract).
2. The source apportionment which is the key focus of this study was done qualitatively because TVR was below 2. A TVR value below 2 signifies that sources are not completely differentiable. In such cases, the uncertainty in the contribution of different sources is higher, which does not mean that an EMMA is useless. I will encourage the authors to undertake a simple EMMA and report the results for the same. An easy way to do this will be using one anion and one cation (reason in #3 below) and some variant of an EMMA. On the point of violation of assumptions, instead of a conventional EMMA, a Bayesian mixing model can be used where the error distribution can be parameterized and later verified.
3. On visual inspection of Figure 4, it seems that $Cl^-$ is markedly different from the other tracers. Most of the cations are positively correlated and offer complementary information. Is this the case? If yes, why not simply use one anion and one cation instead of doing a principal component analysis using all the tracers. The problem with PCA is that readers do not know which tracers influence PC1/PC2 and to what extent, losing physical significance. This will also ensure that an EMMA model can be setup in a very simple way (using one cation and one anion as the tracers)
4. Sections 5.2 and 5.3 can be combined into one section, that will make it easier to read the sections and also help avoid repetitions.

**Minor comments:**
1. The number of sources are different in different parts of this article (eg: P1L15, P3L67, P5L112, P9L232, etc.). I will recommend using the same number of sources at different instances in the article.
2. How many of the 11 snowmelt samples came from North York Creek? (P5L124)

3. Were EC measurements also taken? These can also be used to verify if the seep water is coming from a groundwater pool. (P11L249)
4. Water temperature has been discussed at different places in the article, however there are no figures of water temperature in the article. I will recommend to include at least one figure for water temperature.
5. The reported hillslope groundwater includes riparian water and soil water. How is a riparian zone part of hillslope? (P11L260)
6. Section 5.3 indicates some kind of a hysteresis pattern in the PC plots of streamflow (anticlockwise direction in Star west (Figures 5, 6) and clockwise direction in Star east (Figures 7, 8)). I will encourage more discussion about the reason behind this.
7. There is no work done on the water age, how have old or new water been defined? (P1L18, P18L418, L420, etc.)

---

## Author Comment (AC4) · 29 Jun 2020

**Author response to Anonymous Referee #3: Our replies to referee comments (*black italics*) are provided below in blue.**

*Anonymous Referee #3 comments:*

**Summary of the paper:**
*In this study, multiple tracers were used to identify dominant runoff generation mechanisms over two hydrologic years in Star-east and Star-west watersheds. Principal component analysis was used to reduce the complexity that may arise by analyzing every tracer combination. The study concluded that streamflow during early melt was dominated by hillslope groundwater. As snowmelt peaked, the entire landscape became connected and all the water sources contributed to streamflow (the proportion from different sources is not computed). During the Fall season, hillslope and bedrock groundwater became the major sources of streamflow in Star West watershed (proportions not computed), however the sources were unresolved in the Star East watershed. The au*
*thors then went on to conclude the subsurface flow pathways in this region are complex and this complexity along with slow release of groundwater from glacial till ensures hydrologic resilience in this region.*
*This study tries to resolve the seasonal sources of streamflow which is a very interesting research topic and definitely fit for this journal. However, quantitative estimates of source proportions are missing from this study which is possible to compute given the number of tracer variables that were monitored.*
**Reply:** We thank the referee for their review of this draft manuscript.

**Major comments:**
*1. The abstract and introduction talks in detail about the concept of hydrologic resilience, however I do not find any attempt to quantify this statistic in the remainder of this article (except a very brief discussion on recession rates at the end). I will recommend either quantifying resilience or removing it (at least from the abstract).*
**Reply:** All sections relating to hydrologic resilience will be removed.

*2. The source apportionment which is the key focus of this study was done qualitatively because TVR was below 2. A TVR value below 2 signifies that sources are not completely differentiable. In such cases, the uncertainty in the contribution of different sources is higher, which does not mean that an EMMA is useless. I will encourage the authors to undertake a simple EMMA and report the results for the same. An easy way to do this will be using one anion and one cation (reason in #3 below) and some variant of an EMMA. On the point of violation of assumptions, instead of a conventional EMMA, a Bayesian mixing model can be used where the error distribution can be parameterized and later verified.*
**Reply:** The mixing model portion of EMMA was not run because of the violation of key assumptions, the large variability in source water, and Star East stream water not being bound by its sources. The seasonal variation in stream water and large overall variation in source water added uncertainty to mixing results; median or mean values of source water would not have physical meaning during a given season or month. Additionally, while not mentioned in the draft manuscript, small numbers of samples per source can also add uncertainty to mixing proportions. Small et al. (2002) suggested that greater than 20 samples per source are required to reduce this uncertainty; however, in many cases, we have far fewer than 20 samples. Due to the combination of the factors above, the error associated with the un-mixing model would be very large and results would not be particularly meaningful. We decided that a qualitative description of these data displayed in a PCA would still provide insight into the hydrological processes in our study region because the principal components (PC1/PC2) were created from the variability across multiple tracers.
It is important to note that it has been shown that less accurate predicted mixing proportions can arise from reducing the number of tracers used in the un-mixing model (Barthold et al., 2011). While others have used two tracers historically, close scrutiny of predicted portions using known mixtures have shown that larger number of tracers generate more accurate results (Collins et al., 2017; Sherriff et al., 2015). Thus, undertaking a simple EMMA or a Bayesian mixing model with 2 tracers as suggested would have the same problems regarding source water variation and large uncertainty in predicted proportions compounded with overall less accurate mixing proportions. The importance of testing mixing model

predictions using mixtures, rather than goodness-of-fit tests for the prediction of measured tracer values in mixed waters, as was conventional for many years, has been critical in revealing the dangers of using overly reductionist signatures.

*3. On visual inspection of Figure 4, it seems that Cl- is markedly different from the other tracers. Most of the cations are positively correlated and offer complementary information. Is this the case? If yes, why not simply use one anion and one cation instead of doing a principal component analysis using all the tracers. The problem with PCA is that readers do not know which tracers influence PC1/PC2 and to what extent, losing physical significance. This will also ensure that an EMMA model can be setup in a very simple way (using one cation and one anion as the tracers)*

**Reply:** Yes, most ions are positively correlated but $Cl^-$ is also positively correlated except for a few samples that had higher concentrations. $SO_4^{2-}$ better separates the source and stream water samples along a biplot axis and would likely be the better choice if conducting a 2-tracer mixing space/model. However, as explained above, a 2-tracer mixing model would still have large uncertainties associated with the estimated proportions since overly reductionist signatures generate less accurate proportions versus known mixtures (either virtual or actual). The methods used in this study were intended to maximize the statistical information provided by the tracer suite without overstating the results or conclusions that we could make from this dataset.

To address the problem that readers do not know which tracers influence PC1/PC2, a simple option to help clarify the physical significance of the PCA plot is to include a table with the tracers that influence PC1 and PC2. These types of tables are provided often along with PCA plots and can be added here for clarification.

*4. Sections 5.2 and 5.3 can be combined into one section, that will make it easier to read the sections and also help avoid repetitions.*

**Reply:** Referee #1 suggested we combine sub-sections within Sections 5.2 and 5.3 to avoid repetitions and we agree that this will help streamline the discussion. We will consider combining the source water contributions and stream water contributions discussion sections as suggested by Referee #3. However, we do not want to lose the ability to stress some key discussion points in source water dynamics. Regardless, we will ensure that the Discussion is less repetitive and more streamlined.

***Minor comments:***
*1. The number of sources are different in different parts of this article (eg: P1L15, P3L67, P5L112, P9L232, etc.). I will recommend using the same number of sources at different instances in the article.*
**Reply:** The number of sources in the identified lists will be revised for consistency.

*2. How many of the 11 snowmelt samples came from North York Creek? (P5L124)*
**Reply:** Two of the snowmelt samples came from North York Creek. This will be clarified in the text.

*3. Were EC measurements also taken? These can also be used to verify if the seep water is coming from a groundwater pool. (P11L249)*
**Reply:** EC was taken from seeps and the stream but not the till and bedrock wells, hillslope/riparian wells, or suction lysimeters. We agree that EC could be used to verify the source of the seeps and we will explore these data.

*4. Water temperature has been discussed at different places in the article, however there are no figures of water temperature in the article. I will recommend to include at least one figure for water temperature.*
**Reply:** We will consider adding a figure for water temperature. Referee #1 also had a similar suggestion about groundwater temperatures. Water temperature of seeps were discrete measurements taken during three flow conditions over two seasons. It is likely that a box and whisker plot will be the best way to display these data because only the till and bedrock wells and stream have continuous temperature data. Water temperature was not measured in hillslope/riparian wells.

*5. The reported hillslope groundwater includes riparian water and soil water. How is a riparian zone part of hillslope? (P11L260)*

**Reply:** Hillslope groundwater included riparian water when the chemical signature of riparian water was not statistically different from the chemical signature of soil water. While the processes that occur in the riparian area certainly differ from those on the hillslope, there was not a significant difference between these sources at Star West Lower and Star East Upper so these sources were grouped together.

*6. Section 5.3 indicates some kind of a hysteresis pattern in the PC plots of streamflow (anticlockwise direction in Star west (Figures 5, 6) and clockwise direction in Star east (Figures 7, 8)). I will encourage more discussion about the reason behind this.*

**Reply:** The difference in direction between Star West and Star East is likely a product of the PCA process, not due to a difference in hydrological processes, as the tracer's eigen vectors have opposing signs (positive vs negative) in Star West and Star East. More discussion about the difference in direction of the hysteresis loops will be added for clarification.

*7. There is no work done on the water age, how have old or new water been defined? (P1L18, P18L418, L420, etc.)*

**Reply:** No work has been conducted on water age. Referee #2 has also suggested we clarify the definition of old water to mean "reacted waters". References to old water will be changed to make it clear that we are talking about water that was already in the watershed prior to snowmelt in contrast to new water such as rain and snow.

---

## Author Response (AR1)

**Our replies to referee comments (black italics) are provided below in blue.**

**Author response to Anonymous Referee #1:**

Anonymous Referee #1 comments:

**General comment**

This works uses hydrochemical data to describe and infer runoff generation processes in the subcatchments of the Rocky Mountains. The topic is certainly interesting for the readership of HESS. The manuscript is generally well written. However, there are two main points that do not sound convincing to me: i) the focus on catchment resilience and disturbance, that do not appear to be logically linked to the investigations carried out and sounds out of context; ii) the presence of hydrochemical data only: despite the powerful nature of hydrochemistry as hydrological tracer, the combination of racer data and hydrometric data can help to unravel the complexity of hydrological processes at the catchment scales. Thus, the manuscript fails to describe in a robust, quantitative, and convincing way how water moves through this landscape in response to both rainfall and snowmelt. As a result, a clear contribution of this study to the body of knowledge is not evident. Please, find some specific and minor comments below.

Reply: The authors thank the referee for their comments.

i) Hydrological resilience observed in this region (e.g., Harder et al., 2015; Goodbrand and Anderson, 2016) was the motivation to undertake this research. Others have suggested that complex subsurface flow pathways and large subsurface storage are potential factors that lead to hydrologic resilience (Harder et al., 2015) but, critically, little is known about runoff generation in the eastern slopes of Alberta's Rocky Mountains. To address this evidence gap, developing a conceptualization of groundwater-surface water interactions and runoff generation processes was the first step towards understanding why this region appears to be resilient to change. Despite this, we understand the referee's concerns about the lack of linkage between resilience and the research presented in our draft manuscript. To address the concerns of the referee, the draft Introduction has been reformulated to more clearly draw attention to the poor current understanding of runoff generation in regions with both highly permeable bedrock and deep soils/glacial till in contrast to conceptually less complex and comparatively well studied systems with impermeable bedrock and shallow soils because this was ultimately our intention. We believe this more clearly establishes the impact of this work in advancing the body of knowledge on catchment scale runoff generation. All references to watershed resilience have been removed as suggested.

ii) We agree with the referee's comment regarding coupled geochemical and hydrological data to unravel hydrologic behaviour at the watershed scale. Indeed, this study is part of a larger research project that has been published in part. Our first published manuscript includes estimates of dynamic storage by water balance method, a hydrograph recession analysis to infer groundwater contributions, snowfall-runoff ratio relationships to understand the influence of pre-winter storage, and hydrograph response at the storm scale in wet and dry seasons (Spencer et al. 2019). These analyses were used to develop a conceptual model of storage and runoff generation for alpine and subalpine/montane regions in Star Creek. This was the first step in understanding runoff generation in regions with glacial till overlaying permeable sedimentary bedrock. The aim of this draft manuscript submitted to HESS is to develop further lines of evidence to address the complexity of runoff generation in this region. While Harder et al. (2015) postulated that this region has complex subsurface flow pathways, to our knowledge, no studies have characterized runoff generation in the subalpine region of Alberta's Rocky Mountains. The Introduction of the present draft manuscript has been reformulated to stress the connection with Spencer et al. (2019) and position this research in context with other regions with shallow soils and impermeable bedrock.

It was an oversight not to include any data that shows the runoff patterns in the draft manuscript. Figure 3 new has been added to link observed inputs (daily precipitation and continuous snow depth) to observed responses (specific discharge, stream water chemistry, and shallow groundwater levels). Similar figures are often presented along with hydrochemical data in other publications (e.g., Blumstock et al., 2015; Barthold et al., 2017; Ali et al., 2010; Cowie et al., 2017; Inamdar et al., 2013; Hoeg et al., 2000; Sueker et al., 2000; Correa et al., 2019). A table with precipitation event characteristics is also often presented in conjunction with the hydrographs; however, storms were not the focus of our study so other metrics such as average annual discharge, percent of streamflow that occurs from May to July, and annual peak flow have been added to further characterize the hydrological setting.

We have not included unmixing model results because of the violation of multiple EMMA assumptions. We had explored this option before and decided that the errors associated with the percent contributions outweighed the potential contributions. This is similar to Hoeg et al., (2000) who found geographic hydrograph separations did not lead to conclusive results, and Inamdar et al., (2013), who suggested that caution should be applied when calculating percent contributions where there is large variation in end members and assumptions are violated. If the critical assumptions of EMMA were not violated, we would also have examined model accuracy using virtual mixtures of the sampled water sources since model tests using such mixtures are becoming more commonplace in the international literature using source apportionment methods. However, as we state above, assumption violation precludes inclusion of such work.

**Specific comments from Reviewer 1:**

The abstract is a bit vague. The motivation sounds weak, there are no specific objectives, the methods are partly unclear (water sources were sampled for what kind of analysis?), and the concept of hydrological resilience is not specified. I suggest revising it entirely.

**Reply:** This comment is connected with this referee's general comments above. The abstract has been heavily revised to reflect the changes in the revised Introduction and Discussion and to clarify the conclusions that were made in relation to runoff generation.

- The Introduction fails to clearly stress what it is not well known about the specific topic and what is the main research gap, and the reader, at the end of the Introduction is left wondering why another study on streamflow contribution is needed. An overall objective and testable hypothesis is not reported. The two specific objectives are introduced quite abruptly, without a clear and logical connection with the paragraph above. I suggest to heavily revise the Introduction to keep these points into consideration.

**Reply:** This comment also reflects the referee's general comments above. The draft Introduction has been heavily revised to better clarify research gaps and the concomitant rationale for this study.

- 190-208. I suggest to consider the work by Barthold (2001) and to specify the reported approaches were preferred over this method. Moreover, briefly mention how TVR and LDA work to allow the reader better understanding the methods that were used.

https://agupubs.onlinelibrary.wiley.com/doi/pdf/10.1029/2011WR010604

**Reply:** The draft text has been revised to specify why the approach used here was preferred over the method outlined in Barthold et al. (2011).

"Although others have often used the selection criteria presented in Barthold et al. (2011), an inability to run the unmixing routine (stream water fell outside the bounds of the source water; see Results) hindered the use of these methods. Rather, the tracer variability ratio (TVR; explained below) and linear discriminant analysis (LDA; explained below) have been presented as effective parameters to subjectively determine if tracers are included in the analysis and if sources are well separated or grouped appropriately, respectively (Pulley et al., 2015; Pulley and Collins, 2018; and others – see comprehensive review in Collins et al., 2017)."

The explanation of how TVR and LDA work has been expanded.

"TVR was calculated using the following equation: the percent difference between source group median divided by the average coefficient of variation between group pairs, for each tracer pair and compared between each group (Pulley et al., 2015)."

"LDA optimizes separation between centroid of group clusters by partitioning the variation across each tracer and weighing that variation into two axes (reducing the dimensionality). Other statistical classification methods, such as hierarchical clustering or k-means clustering, were not appropriate because source categories were known a priori. The data were processed in R (R Core Team, 2014) using 'lda' function in the MASS package (Venables and Ripley, 2002) to reduce dimensionality and assess the separation visually and the klaR ('stepwise' function; Weihs et al., 2005) package to model the data and determine the ability to separate groups statistically. The 'stepwise' function models the data while removing individual tracers iteratively. The 'backwards' direction was used in an attempt to maintain the most tracers with the 'lda' method, and 'ability to separate' criterion."

- I suggest merging Figs. 5 and 6 (making a multi-panel figure) and sections 5.2.1 ad 5.2.2, and Fig. 7 and 8 and sections 5.2.3 and 5.2.4 in order to present the results from the two subcatchments more organically. Similarly, I recommend merging Section 5.3.1 and 5.3.2 (Star West), and 5.3.3 and 5.3.4 (Star East) to avoid too much text and results in fragmentation. **Reply:** These sections and figures have been revised as suggested.

- Since there is, at least in some cases, a strong seasonal pattern in hydrochemistry, I suggest considering making a time series plot of the different water sources in the two subcatchments in order to show, for instance, when and to which extent the stream water signature gets closer to that of hillslope groundwater and riparian water. In addition, the Authors might consider adding times series of groundwater temperatures or boxplots, as this tracer is part of the story and was shown to be able to partly explain groundwater contributions to streamflow.

**Reply:** Time series plots of various ion concentrations for water sources and stream water have been added as suggested. However, bedrock groundwater was not included in the time series because those samples were collected in 2016 and 2017 while all other samples were collected in 2014 and 2015. These time series plots have replaced the insets in Figures 7 and 8 because they better illustrate the patterns we describe in the text. In addition, a boxplot of groundwater temperatures, stream water temperatures, and seep temperatures has been added as suggested.

- 417-418. Which evidence do the Authors have to infer the temporal dynamics of hillslope water moving to the stream? Moreover, how could the Authors describe old water mobilization without having quantified its proportion in stream water? Or this is a general statement not based on the presented dataset? Please, explain.

**Reply:** Temporal dynamics of hillslope water moving to the stream are inferred from the PCA plots as stream water is similar to the various source water at different times of the year as well as the time series of streamflow, shallow groundwater and snowmelt (Figure 11). Figure 11 also shows the concentration of calcium increased in the early spring in 2014 just as snowmelt starts and before there is a large response in the stream. Hillslope-stream connectivity was inferred from the shallow water table responses, similar to Jencso et al. (2009). The draft manuscript has been revised to clarify that we are using these lines of evidence to infer the temporal dynamics of hillslope water moving to the stream.

- 464-474. I feel this part is quite out-of-context and disconnected from the previous discussion. In general, I think that focusing on catchment resilience is not so straightforward and sound a bit contrived to me. The same comment applies to the Conclusions.
 Reply: All references to catchment resilience have been removed as suggested.

Minor comments and technical corrections:

*1. The title is long and complex. I suggest making it more compact and clearer.* **Reply:** The title has been revised to: "Hillslope and groundwater contributions to streamflow in a Rocky Mountain watershed underlain by glacial till and fractured sedimentary bedrock". 11. I suggest to change as follows: "A lack of : : :but mechanisms governing: : : ". **Reply:** This sentence was removed during revision of the draft Abstract.

13. ": : : although much: : : ": I cannot see the logical link in this sentence. Please revise. **Reply:** The abstract has been revised for clarification and to reflect the changes in the Introduction and additions to the draft manuscript. This sentence has been removed.

13-14. "to interpret how forest disturbance may impact streamflow quantity". I would not focus on understanding runoff generation processes to this aim, but mostly on the ecohydrological role of forest on streamflow. Please, revise.

**Reply:** The focus of the manuscript has been revised. See reply to general comment and Introduction comment above. The abstract has been revised for clarification and to reflect the changes in the Introduction and additions to the draft manuscript.

**22. "but was unlike the measured sources": this sentence is not clear before reading the abstract. Please, clarify.**

**Reply:** Revised to: "Conversely, in Star East, the composition of stream water was similar to hillslope water in August but plotted outside the boundary of the measured sources in September and October. The chemical composition of groundwater seeps followed the same temporal trend as stream water, but consistently cold temperatures of the seeps suggested deep groundwater was likely the source of this late fall streamflow."

*29: Perhaps put it more general, mentioning pathogens.* **Reply:** Revised as suggested.

35. What do the Authors refer to by "features"? Please explain.

**Reply:** Here, "features" was referring to "watershed features (e.g., bedrock, surficial geology, wetlands)" and has been revised for clarification.

**112. ": : : a priori: : : ": Was there any evidence, field observation, previous study or knowledge of the area that allowed for this assumption?**

**Reply:** Stream water sources were hypothesized based on field observations and previous knowledge of the area. This research is part of the Southern Rockies Watershed Project, which has been conducting research in this watershed and other watersheds in the area since 2004. As such, the stream water sources were based on local knowledge from working in these mountains. This has been clarified in the text.

**193. TVR: please report the definition and possibly the equation to let the reader immediately understand it.**

**Reply:** The text has been re-arranged for clarification. The definition and equation have been stated more explicitly so the reader can immediately understand it.

229-230. This sentence is not clear to me (without reading the cited references). Please specify.
Reply: Revised to: "Greater variation within- compared to between-source groups violates assumption
3) for EMMA (source water does not change) and was considered unacceptable. As a result, rather than calculating mixing ratios or percent contribution of sources to stream water on the basis of an un-mixing routine in EMMA, trends in stream water distribution were described in relation to source water dynamics and runoff processes."

**245. leu?**

Reply: This was a typo and has been corrected.

**269: Perhaps add "compared to bedrock groundwater".**

**Reply:** Sentence has been revised to: "Hillslope groundwater exhibited greater chemical variation across samples (SD of 3.8 and 2.0 for PC1 and PC2, respectively) compared to bedrock groundwater (SD of 2.9 and 4.8 for PC1 and PC2, respectively), but no clear temporal pattern was observed."

*Fig. 6b). Could the Authors perhaps colour-code samples for season (spring, summer, fall)?* **Reply:** Rather than colour-coding, time series have been added to illustrate this pattern. Insets have been removed because time series have made them redundant.

322. Which are these months?

Reply: Text has been revised to: '...months of open-water flow (Apr-Oct)...'

340. Why a source might be missing? Please, explain.

**Reply:** This sentence was removed when Sections 5.3.1 and 5.3.2 were combined.

*393-394: Are groundwater levels available? Their temporal patterns could help understand which feeds which. Perhaps some piezometers could be installed for a follow-up study.*

**Reply:** Groundwater levels are available in the riparian, toe slope and upper hillslope areas. The following sentence has been added: "Timing of riparian and stream water level responses may be used to help clarify these patterns in future research."

429-430. What does "increase in stream water chemistry" mean? Moreover, how would be possible to infer connectivity through hydrochemical data only? Some speculations could be done but a combination of hydrometric and tracer data would serve this purpose better. **Reply:** Revised to "increase in stream water ion concentrations".

These water chemistry observations were taken in conjunction with shallow groundwater wells using similar methods as Jencso et al. (2009) to infer connectivity. Figure 11 new has been added to help visualize connectivity in conjunction with stream discharge response and changes in stream chemistry.

*431. Contributions to what? Please specify.* **Reply:** Revised to: 'Source water contributions to the stream...'

**433. It cannot be all rain water, can it? Please, revise/explain.**

**Reply:** No, this is not all rain water. It is almost entirely snowmelt as this is a snow-dominated watershed but some summer rain storms would also contribute to runoff. Snowmelt saturates the landscape in May and causes a significant dilution effect in the stream. However, the water that is contributing to the stream is not as dilute as snowmelt water itself, suggesting that there is a mixture of snowmelt and hillslope water contributing to the stream. We state that it is most like precipitation to stress this dilution effect, but not to suggest that the water entering the stream is pure snowmelt or rain. Water chemistry of rainfall and snowmelt were essentially identical so there was no way to separate the contribution of rain and snow.

Text added: "It should be noted that although dilution was observed, stream water was less dilute than snowmelt alone. Snowmelt water interacts with the soil as it moves through the subsurface to the stream directly influencing the chemical composition of the snowmelt contributions to streams (Sueker et al., 2000)."

Possible useful readings for additional analyses and for the discussions section:

Correa, A., Breuer, L., Crespo, P., Célleri, R., Feyen, J., Birkel, C., Silva, C., Windhorst, D., 2019. Spatially distributed hydro-chemical data with tempo-rally high-resolution is needed to adequately assess the hydrological functioning of headwater catchments. Science of The Total Environment 651, 1613–1626. https://doi.org/10.1016/j.scitotenv.2018.09.189

Godsey, S.E., Hartmann, J., Kirchner, J.W., 2019. Catchment chemostasis revisited: Water quality responds differently to variations in weather and climate. Hydrological Processes 33, 3056–3069. https://doi.org/10.1002/hyp.13554

*Hoeg, S., Uhlenbrook, S. and Leibundgut, C., 2000. Hydrograph separation in a mountainous catchment - combining hydrochemical and isotopic tracers. Hydrol. Process., 14: 1199-1216. doi:10.1002/(SICI)1099-1085(200005)14:7<1199::AIDHYP35>3.0.CO;2-K*

Hrachowitz, M., Bohte, R., Mul, M.L., Bogaard, T.A., Savenije, H.H.G., Uhlenbrook, S., 2011. On the value of combined event runoff and tracer analysis to improve understanding of catchment functioning in a data-scarce semi-arid area. Hydrol. Earth Syst. Sci. 15, 2007–2024.

Nadal-Romero, E., Khorchani, M., Lasanta, T., García-Ruiz, J.M., 2019. Runoff and Solute Outputs under Different Land Uses: Long-Term Results from a Mediterranean Mountain Experimental Station. Water 11, 976. https://doi.org/10.3390/w11050976

Penna, D., van Meerveld, H.J., Zuecco, G., Dalla Fontana, G., Borga, M., 2016. Hydrological response of an Alpine catchment to rainfall and snowmelt events. Journal of Hydrology 537, 382–397. https://doi.org/10.1016/j.jhydrol.2016.03.040

Suecker, J.K., Ryan, J.N., Kendall, C., Jarrett, R.D., 2000. Determination of hydrologic pathways during snowmelt for alpine/subalpine basins, Rocky Mountain National Park, Colorado. Water Resour. Res. 36, 63–75. https://doi.org/10.1029/1999WR900296

**Reply:** Thank you for these suggestions. These references were incorporated where applicable.

**Author response to Editor comment:**

This reviewer is very critical about the scientific (added) value of the paper and recommends rejection in the formal reviewer report. This overall assessment is motivated by the statement that "the manuscript fails to describe in a robust, quantitative, and convincing way how water moves through this landscape in response to both rainfall and snowmelt.".

This statement is motivated by two main critics:

"i) the focus on catchment resilience and disturbance, that do not appear to be logically linked to the paper investigations carried out and sounds out of context;

*ii) the presence of hydrochemical data only: despite the powerful nature of hydrochemistry as hydrological tracer, the combination of racer data and hydrometric data can help to unravel the complexity of hydrological processes at the catchment scales."*

The authors' response to the second point does not contain any detailed plans at this stage. I am waiting for a third editor report but I already invite the authors to provide a more detailed planned how you plan to address the above second point before I can give a recommendation on the submission of a revised version.

**Reply:** We thank the editor for their comments and clarification on this issue. We have elaborated on our comments and edits to Referee #1 (above).

**Author response to Anonymous Referee #2:**

Anonymous Referee #2 comments:

**Overview**

This is an interesting research manuscript with well-defined objectives of using EMMA to understand relative variation in stream water sources in forested and alpine watersheds of the Canadian Rocky Mountains. Overall the manuscript is sound and has relatively minor grammatical errors. However, it is suggested that the introduction and discussion sections more clearly identify the secondary objectives and how the study. It is unclear if the results of this study have a primary goal of improving understanding of the variability between different groundwater sources of (e.g. bedrock groundwater vs. glacial till groundwater) in alpine and sub-alpine watersheds, or to understand/predict how future forest disturbance will impact watershed hydrology and runoff generation. **Reply:** Firstly, we thank the referee for their positive comments.

This is a similar comment to the general comments by Referee #1 who suggested that any discussion of hydrologic resilience be removed. We agree that revising the draft Introduction will strengthen the context of this research and clarify that the impetus to conduct this study was to improve our understanding of runoff generation in watersheds with permeable fractured bedrock and deep glacial till

(compared to systems with impermeable bedrock and shallow soils that have been more extensively studied). The primary goal was to improve understanding of variability between different groundwater sources, rather than predicting how future forest disturbance will impact watershed hydrology and runoff generation. The draft Introduction and Discussion have been revised for clarification in response to this, and the previous referee's general comments.

It is suggested that the study site description be expanded to more clearly quantify the size and extent of surface and sub-surface biological and geological features (talus slopes, alpine areas, glacial till, forested areas, riparian areas, etc.). It was hard to determine where and why the division of upper and lower sub-watersheds was chosen for this study and how that division was critical to addressing the question of the impacts of forest disturbance on hydrologic disturbance. Both upper and lower sub-watersheds had forested areas so the separation of upper and lower sub-watersheds did not appear to be a proxy for forested vs unforested/disturbed areas.

**Reply:** The separation between upper and lower watersheds was made in 2008 in anticipation of the Mountain Pine Beetle expanding into our research watersheds (which did not actually occur). The upper stations were positioned at the transition between 1) primarily lodgepole pine dominated forests (upper Montane ecozone) and 2) the narrower band of subalpine fir dominated stands and the treeless alpine valley above this (sub-alpine and alpine ecozones). The higher elevation sub-watersheds would have theoretically been spared from the pine beetle attack allowing for a potential comparison after beetle attack if that had actually occurred. Pine beetle did not reach our research watersheds, so they were maintained as reference watersheds. The upper sites were maintained because they represent the difference between subalpine/alpine zone and the upper montane zone.

The upper montane zone is fully forested, and slopes are slightly gentler. The subalpine/alpine zone is partly forested but also has regions of talus slopes, near vertical bedrock cliffs, and alpine grasses. The streams in the alpine flow over bedrock in some places with colluvial or alluvial material present in other areas. Rather than being a comparison of forested and unforested sub-watersheds, the research presented here capitalized on this nested gauging data to advance the conceptualization from Spencer et al. (2019), comparing subalpine/alpine to upper montane responses.

The draft Study Site description has been expanded to clarify the differences between the upper and lower sub-watersheds. Talus slopes, forests, glacial till, and alpine areas have been added to Figure 1 and the area of surficial deposits have been quantified in the text. Riparian areas have not been added because we do not have accurate measurements of riparian widths at this time.

Secondly, the discussion and conclusion sections should be expanded to more clearly indicate how the results in this study advance or improve the scientific understanding of "how forest disturbance may impact streamflow quantity". It is unclear if the study was able to confirm or reject the hypothesis stated in the abstract that "slow release of groundwater from glacial till" (line 24) generates "hydrologic resilience" in the Rocky Mountains?

**Reply:** Again, this is a similar issue as raised by Referee #1 who suggested that all sections relating to hydrologic resilience in the Rocky Mountains be removed. The revised section no longer discusses how forest disturbance may impact streamflow quantity. However, discussion surrounding the slow release

of groundwater from glacial till remains but has been revised to reflect the removal of the aforementioned discussion.

Line 18: Suggest defining "old water" as related to time the water has spent in the watershed rather than the true age of water.

**Reply:** Revised to: "An initial displacement of water stored in the hillslope over winter ("reacted water" rather than "unreacted" snowmelt and rainfall) occurred at the onset of snowmelt before stream discharge responded significantly."

Line 20: In Star east in September and October the stream water was unlike the sources. What is the additional source or is it a mixed signal?

Same statement is in the conclusion but the proposed explanation of the "missing" sources is either absent or not clear in the conclusion. Clarification would be helpful.

Reply: The draft Abstract and Conclusion have been revised for clarification of this issue.

Abstract: "Conversely, in Star East, the composition of stream water was similar to hillslope water in August but plotted outside the boundary of the measured sources in September and October. The chemical composition of groundwater seeps followed the same temporal trend as stream water, but consistently cold temperatures of the seeps suggested deep groundwater was likely the source of this late fall streamflow."

Conclusion: "Star West stream water was once again similar to hillslope water or riparian water, but Star East stream water plotted outside the boundary of the measured sources. Seep water temperatures were cool and had low variability suggesting it may be deeper bedrock water contributing to the stream. Slower recession rates (and likely lower hydraulic conductivity) in the till groundwater well than in the bedrock groundwater well suggest that water recharged into the till groundwater may be slowly released to the stream. Contamination of the till groundwater well made it unclear when it was contributing to the stream but groundwater table fluctuations suggested it is likely contributing during late summer or fall."

Line 29: What is the specific reference for beetle infestation? A few Studies from the Rocky Mountains to consider reviewing, Pugh & Small, 2011. https://doi.org/10.1002/eco.239 Bearup et al., 2014. https://doi.org/10.1038/nclimate2198 **Reply:** Boon (2012) is the specific reference for beetle infestation. The authors incorporated Pugh &

Small (2011) and Bearup et al. (2014) as suggested.

Line 53: Consider reference of Cowie et al. (2017) here as that study does use EMMA to examine potential source waters from bedrock groundwater, glacial till groundwater, talus slope water, and soil water on streamflow contributions in forested and alpine watersheds in the Rocky Mountains. **Reply:** Cowie et al. (2017) was not initially included in this section of the draft Introduction because bedrock in the Colorado Rocky Mountains is mainly granodiorite, rather than permeable sedimentary bedrock and we were trying to make a distinction between bedrock types. However, Cowie et al. (2017) is relevant to the current knowledge and overall discussion in this paper and has been incorporated into the revised Introduction. Line 77: Define area weighted precipitation. Was precipitation measured at multiple elevations? With > 1000m elevation change how much does the total precipitation change over that gradient? One suggestion is use of a hypsometric curve to distribute precipitation over elevation (see Cowie et al., 2017)

**Reply:** Precipitation was measured at nine precipitation gauges in Star Creek and a neighbouring watershed (North York Creek). Spencer et al. (2019) used the thiessen polygon area-weighted method to estimate average watershed precipitation rather than using one particular rain gauge to represent the entire watershed. Both area-weighted and hypsometric approaches are good approximations of average precipitation if rain gauges are distributed across a range of elevations. While our gauges were distributed between 1482 m and 1873 m., vertical headwalls and talus slopes in the alpine basins limited the placement of precipitation gauges above 1900 m and in this case hypsometric-based weighting of precipitation may overestimate precipitation at higher elevations where precipitation inputs to near-vertical surfaces is unclear. Accordingly, we choose to use the well-established thiessen polygon method to spatially weight precipitation from our gauge network to reflect elevation as a key factor driving spatial precipitation (i.e. lapse rates) in our study area.

Text has been revised to: "Average annual precipitation was 720 mm at Star Main (1482 m a.s.l.) and 990 mm at Star Alpine (1732 m a.s.l.; Spencer et al., 2019). The area-weighted average annual precipitation (2005-2018) was 950 mm using the Thiessen-polygon method and nine precipitation gauges at a range of elevations in and surrounding Star Creek; 50-60 % of the precipitation falls in the form of snow (Spencer et al., 2019)."

Line 78: Please cite the precipitation and % snow. Is this from the same study (Spencer et al., 2019) which is cited in the discussion in reference to the sub surface storage capacity of the watersheds? **Reply:** Yes, this is the same data from Spencer et al. (2019), although 2015-2018 were added to the years of record. Spencer et al. (2019) has been cited and sentence has been revised to as stated above.

Line 83: "Talus slopes" Please expand this description to include more information on the relative size of this geographic feature in the upper watersheds. Previous studies of source waters to alpine watersheds in the Rocky Mountains (suggested references listed below) indicate that talus slopes and underlying features can be significant source water areas.

Is there any information or indication of permafrost, ice lenses, or rock glaciers in the alpine talus areas that could provide a unique source water?

*Caine, N, 2010. Recent hydrologic change in a Colorado alpine basin: an indicator of permafrost thaw? https://doi.org/10.3189/172756411795932074*

*Clow, D. W., Schrott, L., Webb, R., Campbell, D. H., Torizzo, A., and Dorblaser, M.: Ground water occurence and contributions to streamflow in an alpine catchment, Colorado Front range, Ground Water, 41, 937–950, 2003.*

Hood, J. L., Roy, J. W., and Hayashi, M.: Importance of groundwater in the water balance of an alpine headwater lake, Geophys. Res. Lett., 33, L13405, doi:10.1029/2006GL026611, 2006.

Roy and Hayashi, 2009. Multiple, distinct groundwater flow systems of a single moraine-talus feature in an alpine watershed. https://doi.org/10.1016/j.jhydrol.2009.04.018

Williams et al., 2006. Geochemistry and source waters of rock glacier outflow, Colorado Front Range.

**https://doi.org/10.1002/ppp.535**

**Reply:** Talus slopes (where present) terminate below the near-vertical headwalls the alpine region and in some cases at the transition between alpine-forested regions of the watershed. Streams or tributary features flowing overland from the talus slopes have not be observed. Snowmelt and rain may be temporarily stored in talus slopes as documented in other Rocky Mountain watersheds (Cowie et al., 2017; Clow et al. 2003; Hood and Hayashi et al., 2015; McClymont et al., 2010), but it is highly likely that this water would infiltrate into the subsurface prior to arriving in the stream, thereby changing the chemical concentrations of this water. Permafrost, ice lenses, or rock glaciers are not present in the alpine talus areas of this region based on the data we have from the Alberta Geologic Society, so it is unlikely that they could serve as a potential unidentified source. The draft Study Site description has been heavily revised to address these comments.

The description of geographic features has been expanded so differences between Star Creek and other Rocky Mountain watersheds (indicated above) has been clarified.

Line 86: Can the amount of glacial till deposits be estimated or quantified for the sub-watersheds? There is no indication of spatial extent beyond description on line 80. It would help the reader to understand the potential storage capacity of the till especially since till water was excluded as a potential source water due to sampling well contamination (line 181). One suggestion is moving the citation on line 444 (AGS, 2004) to section 2 study site description and elaborating on the description of the "spatially heterogenous surficial deposits..." to help describe the watershed(s) in more detail. **Reply:** The description of geologic features has been expanded in the revised Study Site description. Data from the Alberta Geologic Society was used to estimate the extent of glacial till deposits and the geospatial distribution was added to Figure 1. Talus slopes were digitized from an orthoimage in ArcGIS and added to Figure 1. Unfortunately, beyond the spatial extent shown in Figure 1 and on the ground inspection of exposed till (observed clay layers vs. unsorted clay to boulder sized material), little is known about the thickness and composition of the glacial till.

The statement "heterogeneous surficial deposits and geology" is based on the fact that across the 10 km2 watershed, there is variability in surficial deposits (e.g., slightly leached till, cirque till, colluvium), geologic formations (various geologic formations, although they are all primarily composed of shale and sandstone) and distribution of fractures and faults. As indicated above, the draft Study Site description has been heavily revised to address these comments and the comments above.

Line 95: Figure 1. It would be helpful to define tree line (separation of alpine from forested area within the sub-watershed. Important because the paper is framed as a study related to "forest disturbance" so the alpine portion of the study areas should be clearly separated from the forested areas. Also please add the locations of the seeps that were sampled and used as potential end members in *EMMA*.

**Reply:** The extent of the forested area and seep locations have been added to Figure 1.

Line 125: Snowmelt collection methods. Perhaps expand explanation of the snowmelt sample timing in order to reduce known uncertainty of changes in snowmelt chemistry related to timing of the melt.

There is a known ionic pulse at the initiation of snowmelt (see Williams et al., 2009), which can be followed by dilute meltwater.

**Reply:** In general, snowmelt samples were collected at random and opportunistically when our field crew happened to be on site rather than at specific intervals. "The timing of sample collection was based on access to backcountry sites and were taken opportunistically when crews were in the area and were able to observe active snowmelt" has been added to the sampling description to clarify the sampling regime.

Time series of stream and source water have been added to the manuscript to address comments by Referee #1. These time series show the presence of a pulse in ion concentration at the onset of snowmelt in 2014 but not in 2015. Samples collected in early May in 2014 had elevated concentrations of ions compared to samples collected in early June. The pulse of higher concentrated meltwater was likely missed in 2015 due to the very early melt. The known ionic pulse in snowmelt to which Referee #2 is referring is very interesting and has been added to the discussion: "Although three snowmelt samples in 2014 showed similar ionic pulses early in the snowmelt season to those reported in the Colorado Rocky Mountains (Williams et al., 2009), the concentrations were notably less than from all other sources and thus not likely an important source of the observed early season increase in stream water concentration of some ions."

Are there any occurrences of dust on other impurities in the snowpack in this region which could impact the snowmelt chemistry or the timing and magnitude of snowmelt? Dry deposition was mentioned for rain water collection (line 121) but not for snowmelt.

**Reply:** Dry deposition of dust/dirt can be a problem in the summer when the landscape is directly exposed to wind but is not an issue in the winter when the ground is frozen and snow covered. Dry deposition from major cities and industrial areas is not known to be a problem because neither are in proximity to the study site. However, organic material shed and transported by wind from forest vegetation and excreted from wild animals would be deposited onto the snowpack.

**Line 171: Is the data from the Hobo sensors used in this paper? If not then this method does not support the paper and should be removed.**

**Reply:** Data from Hobo sensors were used in Figure 13 and to determine the temperature range of bedrock and till groundwater in wells (Figure 5).

Line 274: Bedrock groundwater, "excluded as a source at the upper sites". Please explain how the groundwater seep used in SEU (line 313, figure 8b) was classified as having consistently cool GW temperatures, but was not considered to be a "bedrock groundwater source"? **Reply:** These sections have been revised for clarification.

"Bedrock groundwater samples were collected from a lower elevation in the watershed and may not be representative of higher elevation groundwater chemical composition; therefore, they were excluded from the analysis for the upper sites. Further, there were only two seeps identified in the upper watershed, but the temperature and chemical composition of these seeps were not reflective of bedrock groundwater. While this did not exclude bedrock groundwater contributions to streamflow in the upper regions of the watershed, it showed the chemical composition of the bedrock well and the two seeps may not have been representative of the bedrock groundwater chemistry in the Star West Upper sub-watershed."

"A single groundwater seep was identified in SEU. The seep was chemically similar to stream water but temperatures were consistently cool and indicative of a deep groundwater source so it was retained to aid in the explanation of stream water dynamics (Figures 8 and 9)."

*Line 276: Suggest replacement of "a couple samples" with a more quantitative description.* **Reply:** Revised to: "…except for four snow samples and one rain sample…"

Figures 5-8: Suggest a more detailed explanation of the hysteresis present in the stream water samples. One option is to place the day of year (DOY) on each sample so readers can decipher movement within months which are plotted as one color. For example in figure 7A are the September samples temporally migrating in the mixing space or are sample points randomly distributed?

**Reply:** We tried adding the Julian day to the figure but there are two years of data in the figure so adding dates created too much clutter and did not contribute to the description of hysteresis. In general, sample points are randomly distributed so categorizing by month is the smallest timestep that we can show in these figures. However, the time series (Figures 7, 9, and 10) that have been added to the revised manuscript help illustrate patterns in the chemical composition of stream and source water at a finer scale.

Detailed explanation of the hysteresis patterns has also been added to the discussion: "Differences in the east and west forks were also evident in the hysteresis pattern in stream water chemistry from spring to fall. Star West sub-watersheds had a counterclockwise pattern, whereas Star East sub-watersheds had a clockwise pattern. In general, this is an artefact of the PCA analysis driven by the specific ions that defined each PC (Table 1). In Star East, the first PCs were dominated by anions and the second PCs were dominated by SO42- (negative relationship). While the first PCs for Star West were dominated by anions, the second PCs included a mix of anions and cations and SO42- with a positive correlation thereby producing an opposite hysteresis pattern. Although this is an artefact of the PCA analysis, it was ultimately due to slight variations in the sources contributing to the streams at different times during the flow season."

Figures 7 and 8: SEU and SEL both appear to have an unidentified source water in October as the October samples plot further away from the identified potential end-members. A more detailed interpretation of this observation is recommended for the discussion? Reply: The discussion surrounding the unidentified source in October has been greatly expanded as suggested. See Paragraph 8 in discussion.

Line 325: Section 5.3: It is understood that you were not able to sample in the winter, however you state that sampling stopped "before fall rains" (line) in previous section and in this section the "end" of seasonal sampling is stated as "start of the next year's snow accumulation period" (line 326). Just want to be clear on the terms used to describe the end of seasonal study periods.

Reply: Groundwater seeps were sampled only three times a year due to the large amount of resources it

required compared to other sources. All other sources (other than snow) were sampled between April and October. The description for groundwater seep sampling has been revised to clarify this difference.

Added text: "This sampling campaign required more resources than for other sources, as a result sampling was completed only three times a year during hydrologically important extreme flow conditions rather than every two weeks from April to October as for other sources."

If precipitation is lumped by rain and snow how do you know which form of precipitation is influencing stream flow in which season? For example line 342, the stream is "more similar to precipitation in June and July" Is this recent precipitation from rain or assumed to be the lagged input of snowmelt from the previous winter?

What would be helpful is a hyetograph over the study period so reader has some better sense of when the annual precipitation occurs. Also is there a way to present the timing and magnitude of snowmelt? Figure 10 suggests that there are multiple snowmelt pulses in winter and spring, can this be elaborated in the description of site climate and hydrologic inputs?

**Reply:** Ion concentrations of rainfall and snowmelt were essentially identical so there is no way to distinguish between these two sources based on the chemistry we have (we do not have isotopes). Thus, the lagged input of snowmelt and recent rainfall inputs cannot be separated. Further, the 2-week sampling schedule did not allow for the resolution needed to really identify a rainfall pulse moving through the watershed. Figure 11 (new) has been added to better describe the relationships between when snowmelt was occurring (snow depth time series), the distribution of daily precipitation (snow vs rain), shallow water table responses, and the annual hydrograph. Stream water chemistry has also been included to show how snowmelt dilutes stream water chemistry over the melt period and the recovery (increase) in concentration later in the summer. Continuous snow depth measurements are used in lieu of snow pillow data (snow water equivalent) because SWE was not available. Thus, while we cannot describe how much water is being lost from the snowpack (change in SWE), we can describe the timing of melt.

Line 399: "increases the concentration in water" should be "increases tracer concentrations in the soil water.." if you are speaking about the inverse of water chemistry "dilution" from snowmelt. **Reply:** Revised to: "...thereby increasing ion concentrations in the soil water ..."

Line 429: Please clarify "increases in stream water chemistry" to specify that you are speaking about tracer concentrations or "concentration of stream water ions" (line 450). Consistent terminology will help the flow of the manuscript.

**Reply:** Revised to: "...an increase in stream water ion concentrations..."

Line 457: Please provide citation for this statement "Excess water associated with forest disturbance would infiltrate into the subsurface". These assumed hydrologic dynamics should be discussed in more detail because there is potential for a varying hydrologic response from forest disturbance. For example, in a forested snowmelt dominated watershed the timing and magnitude of snowpack accumulation and ablation in relation to canopy cover/density dynamics may be variable depending on forest dynamics. Sublimation rates on canopy snow interception (see Classen and Downy, 1995), and impacts of forest shading on radiative forcing on snowpack ablation could influence infiltration rates. I would also suggest mention of rainfall intensity relative to infiltration capacity in forested vs alpine or disturbed areas. Recommended references to review:

Molotch et al., 2009. Ecohydrological controls on snowmelt partitioning in mixed-conifer sub-alpine forests. https://doi.org/10.1002/eco.48

Harpold et al., 2014. Soil Moisture response to snowmelt timing in mixed-conifer subalpine forests. https://doi.org/10.1002/hyp.10400

Musselman et al., 2012 Influence of canopy structure and direct beam solar irradiance on snowmelt rates in a mixed conifer forest. https://doi.org/10.1016/j.agrformet.2012.03.011

**Reply:** This statement and the rest of the discussion on hydrologic resilience has been removed as suggested by Referee #1.

Line 475: Figure 10 caption revision. Second sentence is an interpretation of the graph rather than a description and should be included in the text. Recommend clarifying text description of "more responsive" and "slower recession slopes" in reference to depth to groundwater below the surface. **Reply:** The second sentence in the figure caption has been removed as suggested. The interpretation described in body of the text: "The till groundwater well showed consistently slower recession curves compared to the bedrock groundwater well" has been expanded and revised to:

"Water table depth in the till groundwater was more responsive than bedrock groundwater level in the spring, though the overall rise in water level in the bedrock was slightly greater. Despite the flashier response earlier in the year, till groundwater levels remained elevated longer than bedrock groundwater resulting in a slower recession (slower drainage) in the till groundwater well in the summer (Figure 13)."

Figure 10: In the soil/till GW, what causes the sharp response (increase in water table elevation) in November? Is this related to early season snowfall that melts or other factor such as vegetative senescence? Does the chemistry change in that water source in late fall?

Can you explain the two separate groundwater level increases in the till well that occur in February and then again in March/April? Is this related to intermittent snowpack throughout the winter (as briefly mentioned in the snowmelt sampling methods line 125)?

**Reply:** Figure 13 (formerly Figure 10) was added simply to characterize the water table recession and infer the hydraulic conductivity of the bedrock well compared to the glacial till well. The fine-resolution temporal responses of wells in November, February, and March/April were not investigated because these responses were for 2017 and we were focusing on 2014/2015 seasons. However, the March response corresponds to the onset of the spring melt season, particularly at the lower elevations where the till well is located. The exact timing of the spring response cannot be directly linked to the responses observed in 2014/2015 because the 2017 season likely had different timing for snowmelt and groundwater table responses. The time scale on the figure has been changed to March-October to reflect a similar time period as stream water and source water samples. March was included in the figure to show the earlier melt response in 2017.

Although the 2017 season is outside the scope of this manuscript, we did investigate the timing of the February and November till well responses in comparison to precipitation timing and air temperature during revision of this manuscript. The till groundwater table response in November 2017 was caused by brief warm air temperatures and a large rainfall event which are common in the late fall/early winter in this watershed. The mechanisms that lead to the February 2017 response are less clear. A large snowstorm occurred in 1.5 weeks prior the well response during a period of very cold temperatures (-15 to -25 °C). Following the snowstorm and in the days leading up to the well response, the air temperature increased significantly to 5 °C and the snow depth decreased. Thus, it is possible that a mid-winter melt event caused the water table response. Again, these events are outside the scope of the current manuscript but can be addressed in a future manuscript.

Line 480: Replace "old water" with a more accurate description representative of transit time or subsurface residence time rather than speaking to the age of the water, or define old water to mean "reacted" waters that have had extended contact time with the sub-surface (see Liu et al. 2004) The same suggestion was made previously for defining the use of "old water" in the abstract. **Reply:** Any reference to "old water" has been revised to "reacted" water or water that was stored in the watershed over winter rather than a specific age of the water.

Line 485: Indicates that till groundwater could be slowly released to the stream (longer recession in Figure 10). It is not clear if the intention was to suggest that this could be the unidentified source water end member in late fall in Star East, but was not was not captured or used in EMMA due to experimental design issues leading to well contamination? Reply: The conclusion has been expanded and clarified:

"Star West stream water was once again similar to hillslope water or riparian water, but Star East stream water plotted outside the boundary of the measured sources. Seep water temperatures were cool and had low variability suggesting it may be deeper bedrock water contributing to the stream. Slower recession rates (and likely lower hydraulic conductivity) in the till groundwater well than in the bedrock groundwater well suggest that water recharged into the till groundwater may be slowly released to the stream. Contamination of the till groundwater well made it unclear when it was contributing to the stream but groundwater table fluctuations suggested it is likely contributing during late summer or fall. More research on the variability of bedrock and till groundwater chemistry is needed to clarify the difference between these sources and their contributions to streamflow throughout the year."

Line 486: Please expand the conclusion/suggestion that till groundwater (although not used as an end member for EMMA) has the potential to mute the effects of disturbance on peak flow. I assume you are referring to forest disturbance, but it is not clear of the locational relationship between till groundwater sources and forested areas within the watersheds. Is the till groundwater believed to be sourced from direct overhead recharge (in the same location as currently existing forests)? or is there another hypothesized mechanism of recharge such as mountain block recharge from higher alpine regions already void of forest cover?

Reply: Reference to resilience has been removed from the conclusions (and the rest of the manuscript)

as suggested by Referee #1. However, clarification of till groundwater responses/sources in Line 486 have been added to the manuscript as indicted above (Line 485).

**Author response to Anonymous Referee #3:**

Anonymous Referee #3 comments:

**Summary of the paper:**

In this study, multiple tracers were used to identify dominant runoff generation mechanisms over two hydrologic years in Star-east and Star-west watersheds. Principal component analysis was used to reduce the complexity that may arise by analyzing every tracer combination. The study concluded that streamflow during early melt was dominated by hillslope groundwater. As snowmelt peaked, the entire landscape became connected and all the water sources contributed to streamflow (the proportion from different sources is not computed). During the Fall season, hillslope and bedrock groundwater became the major sources of streamflow in Star West watershed (proportions not computed), however the sources were unresolved in the Star East watershed. The authors then went on to conclude the subsurface flow pathways in this region are complex and this complexity along with slow release of groundwater from glacial till ensures hydrologic resilience in this region.

This study tries to resolve the seasonal sources of streamflow which is a very interesting research topic and definitely fit for this journal. However, quantitative estimates of source proportions are missing from this study which is possible to compute given the number of tracer variables that were monitored. **Reply:** We thank the referee for their review of this draft manuscript.

**Major comments:**

1. The abstract and introduction talks in detail about the concept of hydrologic resilience, however I do not find any attempt to quantify this statistic in the remainder of this article (except a very brief discussion on recession rates at the end). I will recommend either quantifying resilience or removing it (at least from the abstract).

Reply: All sections relating to hydrologic resilience have been removed.

2. The source apportionment which is the key focus of this study was done qualitatively because TVR was below 2. A TVR value below 2 signifies that sources are not completely differentiable. In such cases, the uncertainty in the contribution of different sources is higher, which does not mean that an EMMA is useless. I will encourage the authors to undertake a simple EMMA and report the results for the same. An easy way to do this will be using one anion and one cation (reason in #3 below) and some variant of an EMMA. On the point of violation of assumptions, instead of a conventional EMMA, a Bayesian mixing model can be used where the error distribution can be parameterized and later verified.

**Reply:** The mixing model portion of EMMA was not run because of the violation of key assumptions, the large variability in source water, and Star East stream water not being bound by its sources. The seasonal variation in stream water and large overall variation in source water added uncertainty to

mixing results; median or mean values of source water would not have physical meaning during a given season or month. Additionally, while not mentioned in the draft manuscript, small numbers of samples per source can also add uncertainty to mixing proportions. Small et al. (2002) suggested that greater than 20 samples per source are required to reduce this uncertainty; however, in many cases, we have far fewer than 20 samples. Due to the combination of the factors above, the error associated with the unmixing model would be very large and results would not be particularly meaningful. We decided that a qualitative description of these data displayed in a PCA plot would still provide insight into the hydrological processes in our study region because the principal components (PC1/PC2) were created from the variability across multiple tracers.

It is important to note that it has been shown that less accurate predicted mixing proportions can arise from reducing the number of tracers used in the un-mixing model (Barthold et al., 2011). While others have historically used two tracers, close scrutiny of predicted portions using known mixtures have shown that larger number of tracers generate more accurate results (Collins et al., 2017; Sherriff et al., 2015). The importance of testing mixing model predictions using mixtures, rather than goodness-of-fit tests for the prediction of measured tracer values in mixed waters, as was conventional for many years, has been critical in revealing the dangers of using overly reductionist signatures. Thus, undertaking a simple EMMA or a Bayesian mixing model with 2 tracers as suggested would have the same problems regarding source water variation and large uncertainty in predicted proportions compounded with overall less accurate mixing proportions. As a result, we did not to pursue this approach in the revised manuscript.

3. On visual inspection of Figure 4, it seems that Cl- is markedly different from the other tracers. Most of the cations are positively correlated and offer complementary information. Is this the case? If yes, why not simply use one anion and one cation instead of doing a principal component analysis using all the tracers. The problem with PCA is that readers do not know which tracers influence PC1/PC2 and to what extent, losing physical significance. This will also ensure that an EMMA model can be setup in a very simple way (using one cation and one anion as the tracers)

**Reply:** Yes, most ions were positively correlated but Cl- was also positively correlated with these same ions except for a few samples that had higher concentrations.  $SO_4^{2-}$  better separated the source and stream water samples along a biplot axis and would likely be the better choice if conducting a 2-tracer mixing space/model. However, as explained above, a 2-tracer mixing model would still have large uncertainties associated with the estimated proportions since overly reductionist signatures generate less accurate proportions versus known mixtures (either virtual or actual). The methods used in this study were intended to maximize the statistical information provided by the tracer suite without overstating the inferences or conclusions that we could draw from this dataset.

Regarding which tracers influence PC1/PC2, a simple option to help clarify the physical significance of the PCA plot is to include a table with the tracers that influence PC1 and PC2 (Table 2). These types of tables are provided often along with PCA plots and has been added to the revised manuscript (Table 1).

4. Sections 5.2 and 5.3 can be combined into one section, that will make it easier to read the sections and also help avoid repetitions.

**Reply:** Referee #1 suggested we combine sub-sections within Sections 5.2 and 5.3 to avoid repetitions and we agree that this will help streamline the draft Results. We have revised these sections to avoid repetition but have maintained separate sections because we did not want to lose the ability to stress some key discussion points in source water dynamics. However, the draft Discussion has been heavily revised to combine Sections 6.1 and 6.2 to add better flow to the revised Discussion and reduce repetition throughout the manuscript.

**Minor comments:**

1. The number of sources are different in different parts of this article (eg: P1L15, P3L67, P5L112, P9L232, etc.). I will recommend using the same number of sources at different instances in the article. **Reply:** These lists have been revised to be consistent across lists. However, upper and lower sites have different numbers of sources based on the outcome of the analysis.

**2. How many of the 11 snowmelt samples came from North York Creek? (P5L124)**

**Reply:** Two of the snowmelt samples came from North York Creek. Text has been revised to: "Nine snowmelt samples were collected from sub-alpine regions of Star Creek and two from North York Creek..."

**3. Were EC measurements also taken? These can also be used to verify if the seep water is coming from a groundwater pool. (P11L249)**

**Reply:** EC was taken from seeps, stream, and till and bedrock wells, but not hillslope/riparian wells or suction lysimeters. EC has been added (Figure 5) to the revised manuscript.

4. Water temperature has been discussed at different places in the article, however there are no figures of water temperature in the article. I will recommend to include at least one figure for water temperature.

**Reply:** A box and whisker plot has been added (Figure 5) to display the range in water temperature in till and bedrock wells, seeps, and the stream. Water temperature was not measured in hillslope/riparian wells.

**5. The reported hillslope groundwater includes riparian water and soil water. How is a riparian zone part of hillslope? (P11L260)**

**Reply:** Hillslope groundwater included riparian water when the chemical signature of riparian water was not statistically different from the chemical signature of water from hillslope wells. While the processes that occur in the riparian area certainly differ from those on the hillslope, there was not a significant difference between these sources at Star West Lower and Star East Upper so these sources were grouped together. The following text has been added for clarification:

"Although riparian water mixes with stream water and should be chemically different from hillslope water as a source, soil water, toe slope water, and upper hillslope water were grouped with riparian water for most sites because the distribution of these samples were too similar to be considered separate

sources. The exception was SEL and SWU in which riparian water was considered as a separate source."

**6. Section 5.3 indicates some kind of a hysteresis pattern in the PC plots of streamflow (anticlockwise direction in Star west (Figures 5, 6) and clockwise direction in Star east (Figures 7, 8)). I will encourage more discussion about the reason behind this.**

**Reply:** The following text has been added as well as Table 1 indicating the ions that define PC1 and PC2: "Differences in the east and west forks were also evident in the hysteresis pattern in stream water chemistry from spring to fall. Star West sub-watersheds had a counterclockwise pattern, whereas Star East sub-watersheds had a clockwise pattern. In general, this is an artefact of the PCA analysis driven by the specific ions that defined each PC (Table 1). In Star East, the first PCs were dominated by anions and the second PCs were dominated by SO42- (negative relationship). While the first PCs for Star West were dominated by anions, the second PCs included a mix of anions and cations and SO42- with a positive correlation thereby producing an opposite hysteresis pattern. Although this is an artefact of the PCA analysis, it was ultimately due to slight variations in the sources contributing to the streams at different times during the flow season."

**7. There is no work done on the water age, how have old or new water been defined? (P1L18, P18L418, L420, etc.)**

**Reply:** No work has been conducted on water age. "Old water" was used in contrast to "newer water" (precipitation) that would be added during snowmelt. Referee #2 has suggested we clarify the definition of old water to mean "reacted waters", water that has spent time in the watershed and reacted with its surroundings. References to old water have been changed to make it clear that we are talking about water that was already in the watershed prior to snowmelt in contrast to unreacted water such as rain and snow.

**List of Relevant Changes:**

- Introduction and context of manuscript heavily revised
- Added box plots of water temperature and conductivity of bedrock well, groundwater seeps, and stream water
- Added time series of relevant ion concentrations
- Added hydrometric data: 1) Table with streamflow and precipitation metrics and 2) Plot with observed inputs (snow depth and daily precipitation) and outputs (stream discharge, Ca concentration, shallow groundwater table)
- Discussion was streamlined into one section to avoid repetition
- Abstract and Conclusion revised to reflect changes
- Many smaller changes made to the manuscript to address Editor and Referees' comments

[revised manuscript text omitted]
 uUnsorted and uncompacted glacial till is generally less than 3 m deep, on average.

with some elay-rich layers distributed unevenly throughout the watershed with an estimated total area of 2.4 km2 (AGS, 2004;

- 130 Figure 1). Some clay-rich till layers, likely from localized glacial ice melt features, occur intermittently throughout the watershed resulting in heterogeneous and uneven distribution of glacial till throughout the watershed. Sedimentary geologic formations (Upper Paleozoic formation, Belly River-St. Mary Succession, and Alberta Group formation) are primarily composed of shale and sandstone (AGS, 2004) and are highly fractured due to folding and faulting (Waterline Resources, Inc., 2013)-(AGS, 2004).
- 135 There areStar Creek includes two main sub-watersheds, Star East (3.9 km2; 1537-2628 m above sea level) and Star West (4.6 km2; 1540-2516 m above sea level). Unvegetated talus slopes (0.50 km2 in Star East and 0.53 km2 in Star West, digitized from orthoimages) and beneath exposed bedrock form the upper portion of alpine zones in both sub-watersheds (Figures 1 and 2). Talus slopes terminate in the alpine and transitional forested regions of the watershed but streams or tributary features flowing from talus slopes have not been observed. There is also no evidence of permafrost, ice lenses, or rock glaciers, unlike in other
- 140 Rocky Mountain regions (Cowie et al., 2017; Clow et al. 2003; Hood and Hayashi et al., 2015; McClymont et al., 2010). Star West has a larger alpine region with cirque till deposits (estimated area of 0.14 km2 (AGS, 2004)) that includes a narrow marshy area proximal to the stream that holds water throughout the summer and drains into the main channel that is primarily bedrock in the upper reaches. The Star East alpine region is smaller and more constricted than in Star West (Figure 2) and is comprised mostly of a grassy meadow with the stream originating from springs where the water table reaches the soil surface
- 145 and is incised in colluvium with large boulders. In the lower reaches, streams in both sub-watersheds are composed of a series of step-pools incised in alluvium and colluvium with some areas of exposed bedrock. where the lower sub-alpine and upper Montane zones are dominated by sub-alpine fir (*Abies lasiocarpa*) and Englemann spruce (*Picea Englemannii*) above lodgepole pine (*Pinus contorta*) dominated forests in lower reaches (Dixon et al., 2014; Silins et al., 2009).
- Two historical streamflow gauging sites exist in each sub-watershed a lower site (Star West Lower (SWL) and Star East Lower (SEL)) near the confluence of the two sub-watersheds (1540 m above sea level) and an upper site (Star West Upper (SWU) and Star East Upper (SEU)) located at approximately 1690 m above sea level in the alpine/sub-alpine transition zone (Figure 1).- The sub-alpine and upper Montane zones are dominated by sub-alpine fir (*Abies lasiocarpa*) and Englemann spruce (*Picea Englemannii*) above lodgepole pine (*Pinus contorta*) dominated forests at lower elevations (Dixon et al., 2014; Silins et al., 2009). Vegetation in upper and lower watersheds (Figure 1) are distinguished by a transition between higher elevation
- 155 alpine heath/shrub vegetation and sub-alpine fir dominated forests in the upper watersheds and lodgepole pine dominated forest in the lower watersheds.

---

## Author Response (AR2)

**Authors Response to Referee 1:**
Referee comments are in *italics*, Author responses are in blue.

General comment
*The manuscript has been substantially revised, implementing my comments and the ones by the other reviewers, and I believe that this extra work by the authors has significantly improved the manuscript.*
*I have only a few minor comments listed below.*

Minor comments and technical corrections
*10. Inferences on what? Please specify.*

Reply: Revised to "While some inferences on the storage and release of water can be…"

*44. Give examples and references.*

Reply: Examples and references added to text. "(e.g., old water contributions to streamflow, macropore flow and complex subsurface streamflow generation (McGlynn et al., 2002); fill and spill hypothesis (Tromp-van Meerveld and McDonnell, 2006); hillslope-stream connectivity (Jencso et al., 2009))."

*72. I'm not a native speaker but I think that "have" should be "has".*

Reply: Revised as suggested.

*178. Can you include one/two more recent references here? I'm sure there are some.*

Reply: Reference added "Williams et al., 2009"

*201-205. Are seeps mostly active during snowmelt or after rainy events? Do they reflect stream dynamics or their discharge is relatively stable through the snow-free period? I think that a slightly more detailed discussion on the representativeness of these seeps is important to corroborate the results.*

Reply: Elaborated in text. "25 visible seeps were identified which ranged in duration and magnitude of their contributions to streamflow. Some seeps were only active during the snowmelt season and recession period, reflecting streamflow dynamics. Other seeps were relatively stable throughout the entire snow-free period or throughout the winter baseflow period."

*240-246. I appreciate that this part has been considered and included in this version of the manuscript. However, I suggest considering moving these lines in the Discussion and focus here more on what you have done that on what you have not done and why. This will streamline this section avoiding pre-introducing, without really explaining, methods such as TVR and LDA.*

Reply: This section has been moved to the Discussion and revised to:

"The inability to run the unmixing routine (stream water fell outside the bounds of the source water) also hindered the use of some tracer selection methods. Other studies have often used the selection criteria presented in Barthold et al. (2011) but the unmixing routine is required for this method. Rather, the TVR and LDA have been presented as effective parameters to subjectively determine if tracers are included in the analysis and if sources are well separated or grouped appropriately, respectively (Pulley et al., 2015; Pulley and Collins, 2018; and others – see comprehensive review in Collins et al., 2017)."

*252-255. This is a weird way to report an equation. Please, write in in mathematical terms.*

Reply: Revised as suggested. "TVR was calculated using the following equation for each tracer and compared between each source group pair:

$$\frac{\frac{\tilde{x}_{max} - \tilde{x}_{min}}{\tilde{x}_{min}} \, x \, 100}{mean\,(CV_{source\;1},\; CV_{source\;2})}$$

where $\tilde{x}_{max}$ is the maximum median tracer concentration of either source group, $\tilde{x}_{min}$ is the minimum median tracer concentration of either source group, and $CV$ is the coefficient of variation (Pulley et al., 2015; Pulley and Collins, 2018).

*287. Please add the p-value.*

Reply: Revised to "(Pearson's r > 0.5, $p < 0.05$)"

*338. I suggest replacing "separation" with "difference".*

Reply: Revised as suggested.

*351 but also 404 and 422. These titles are identical and potentially confusing. Please, elaborate them better putting them in the context of each section.*

Reply: Revised to reflect "source water" or "stream water" sub-sections.

*373 (caption of Fig. 4). "R" should be "r" (also for consistency with line 287).*

Reply: Revised as suggested.

*Table 2. I suggest including average discharge with its standard deviation for both years and both subcatchments.*

Reply: Revised as suggested.

*457. CV of which parameter? Please, specify.*

Revised to "The CV of source water tracer concentrations…"

*Fig. 12. Use more contrasting colours for the symbols.*

Reply: Revised as suggested.